# *Saurodesmus robertsoni* Seeley 1891—The oldest Scottish cynodont

**Tomasz Szczygielski**[1]*, **Marc Johan Van den Brandt**[2], **Leandro Gaetano**[2,3], **Dawid Dróżdż**[4]

**1** Institute of Paleobiology, Polish Academy of Sciences, Warsaw, Poland, **2** Evolutionary Studies Institute (ESI), University of the Witwatersrand, Johannesburg, South Africa, **3** Instituto de Estudios Andinos "Don Pablo Groeber" (IDEAN, UBA-CONICET), Ciudad Autónoma de Buenos Aires, Argentina, **4** Nalecz Institute of Biocybernetics and Biomedical Engineering, Polish Academy of Sciences, Warsaw, Poland

\* t.szczygielski@twarda.pan.pl

## Abstract

Predating Darwin's theory of evolution, the holotype of *Saurodesmus robertsoni* is a long-standing enigma. Found at the beginning of 1840s, the specimen is a damaged stylopodial bone over decades variably assigned to turtles, archosaurs, parareptiles, or synapsids, and currently nearly forgotten. We redescribe and re-assess that curious specimen as a femur and consider *Saurodesmus robertsoni* as a valid taxon of a derived cynodont (?Tritylodontidae). It shares with probainognathians more derived than *Prozostrodon* a mainly medially oriented lesser trochanter and with the clade reuniting tritylodontids, brasilodontids, and mammaliaforms (but excluding tritheledontids) the presence of a projected femoral head, offset from the long axis of the femoral shaft; a thin, plate-like greater trochanter; a distinct dorsal eminence proximal to the medial (tibial) condyle located close to the level of the long axis of the femoral shaft and almost in the middle of the width of the distal expansion; and a pocket-like fossa proximally to the medial (tibial) condyle. *Saurodesmus robertsoni* is most similar to tritylodontids, sharing at least with some forms: the relative mediolateral expansion of the proximal and distal regions of the femur, the general shape and development of the greater trochanter, the presence of a faint intertrochanteric crest separating the shallow intertrochanteric and adductor fossae, and the general outline of the distal region as observed dorsally and distally. This makes *Saurodesmus robertsoni* the first Triassic cynodont from Scotland and, possibly, one of the earliest representatives of tritylodontids and one of the latest non-mammaliaform cynodonts worldwide. Moreover, it highlights the need for revisiting historical problematic specimens, the identification of which could have been previously hampered by the lack of adequate comparative materials in the past.

## Introduction

*Saurodesmus robertsoni* Seeley, 1891 [1] is an enigmatic taxon based on a single problematic limb bone from the Late Triassic of Linksfield, Elgin, Scotland (NHMUK PV OR 28877). Despite its long, relatively rich and regular record of mentions in the scientific literature, its

**Data Availability Statement:** All relevant data are within the manuscript. The specimens described belong to their respective collections held at scientific institutions and museums (see

Institutional abbreviations) and are available for study.

**Funding:** The study was supported by the National Science Centre, Poland (Narodowe Centrum Nauki, https://www.ncn.gov.pl/en) grants no. 2020/39/B/NZ8/01074 (studies and digitization of pantestudinate postcrania) and 2022/47/B/NZ8/01912 (studies and digitization of synapsid postcrania) awarded to T. Sz., and GENUS (DSI-NRF Centre of Excellence in Palaeosciences, UID 86073, https://www.genus.africa) and the Millenium Trust South Africa (no number, https://www.mtrust.co.za) awarded to M. J.V.d.B. The funders did not play any role in the study design, data collection and analysis, decision to publish, or preparation of the manuscript.

**Competing interests:** The authors have declared that no competing interests exist.

systematic history is exceptionally convoluted. The first, never published illustration of the specimen, a watercolour by Jonathan Stiven (~1799–1872), was created in 1840 (Fig 1). Based on the associated caption, NHMUK PV OR 28877 was found by Alexander Robertson. The watercolour is annotated in pencil with an apparently erroneous reference to one of two possible papers by Malcolmson [2, 3], neither of which, however, clearly refers to this particular bone (see Material and methods).

The specimen was first officially announced by Owen [4] in his report on British fossil reptiles for the year 1841 as a turtle femur resembling (but not identical with) that of *Trionyx* spp. Duff ([5], pl. V, Fig 10) presented the first published illustration of the specimen, also captioned as a turtle femur. Anderson and Anderson ([6], footnote by A. Robertson) and Pictet [7] listed it as a femur of *Trionyx* sp. After acquisition by the British Museum (Natural History) in 1854, it was catalogued by W. Davies as a "humerus? of a Chelonian?" [1] (also in the collection inventory book). Jones [8] mentioned it as a femur of a "chelonian animal" and Maack [9] questioned its turtle affinities, but Judd [10] repeated the earlier identification as a femur of *Trionyx* sp. Seeley [11] suggested a dinosaurian affinity for the specimen. Quenstedt [12] denied its identification as *Trionyx* sp. which he attributed to Owen [4], despite the latter only mentioned the resemblance of the specimen to that genus, and probably even as a turtle at all, deeming it "insufficient". Lydekker [13] mentioned it as a right humerus or femur of a turtle referable to '*Chelytherium obscurum*' Meyer, 1863 [14] (synonym of *Proterochersis robusta* Fraas, 1913 [15] (see [16]; application for conservation of the junior name submitted [17]) from the Norian of Germany or a related taxon but gave no justification and noted that it is

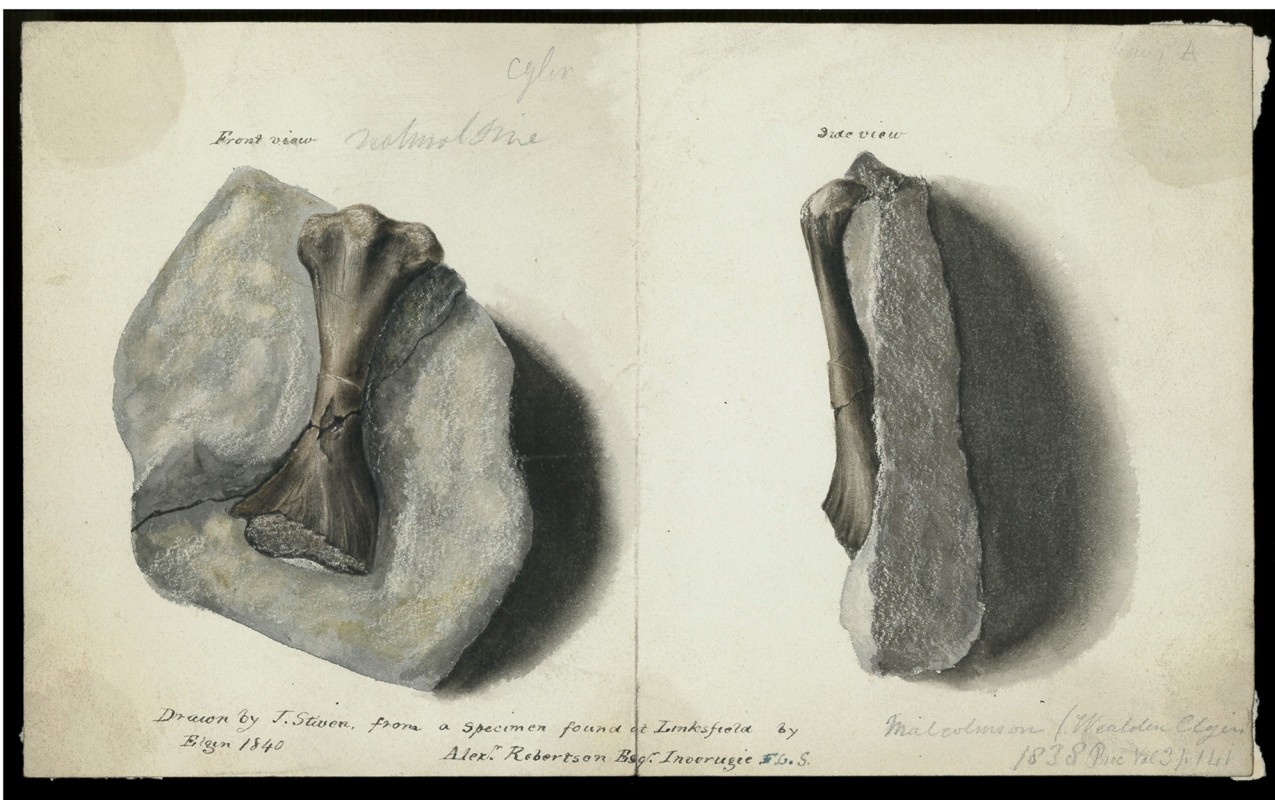

**Fig 1.** *Saurodersmus robertsoni* **NHMUK PV OR 28877, right femur and associated rock fragment before they were separated.** "Drawn by J. Stiven, from a specimen found at Linksfield by Alex^r. Robertson Esq^r. Inverugie F. G. S. Elgin 1840". Watercolour. Note the bone is figured upside-down (proximal end towards the bottom of the page). Reproduced by permission of the Geological Society of London.

different from humeri and femora of any other known turtle. Woodward and Sherborn [18], however, listed it again as a possible femur of *Trionyx* sp. A year later, Seeley [1, 19] revised his earlier interpretation, hesitantly identifying the specimen as a crocodilian right humerus, and was the first to describe the specimen in detail and assign a specific name to it. He also provided more detailed figures, for the first time revealing the ventral surface of the bone, which was embedded in the rock matrix until the preparatory work performed in 1889 [1]. The crocodilian interpretation of the specimen was straight away categorically rejected by Richard Lydekker in the discussion published alongside the paper [1], who suggested once again a testudinate or rhynchocephalian affinity, and the dubious taxonomical identity of the specimen was stressed even in the short synopsis of the paper published the same year in American Geologist [20]. Soon after, Seeley observed some similarity of the proximal end of NHMUK PV OR 28877 to pareiasaurian femora [21] and of the distal end to dinocephalian femora [22–24], allowing the possibility that *Saurodesmus robertsoni* may be an anomodont (note that the group at the time differed in its taxonomic definition from the current understanding of the Anomodontia, as defined by Kammerer & Angielczyk [25]). The perceived similarity with dinocephalians was later reiterated by Seeley [26] in regard to the whole bone. Finally, Seeley [27], revised his interpretation yet again, noticing the resemblance of the proximal part of the femur to the newly established cynodont *Cynognathus crateronotus* Seeley, 1895 [27]; however, he did not specify whether he considered this similarity systematically important. Huene [28] classified the specimen as a right crurotarsan (his Parasuchia, but in contrast to the modern understanding of the group, including a mix of phytosaurs and aetosaurs) humerus, but noted [28] that it may represent his Lycosauria (a group within Theriodontia in Anomodontia including several other problematic taxa). Subsequently, he mentioned the specimen as a crocodile humerus [29]. Nopcsa [30], on the contrary, refuted the identification of the bone as an archosaurian humerus based on the morphology of the proximal end and as a non-archosaurian reptilian humerus based on the lack of the ectepicondylar foramen. Instead, he considered it to be a pleurodiran turtle femur. Curiously, NHMUK PV OR 28877 was listed by Hummel [31] in the fossil trionychian part of "Fossilium catalogus" as a femur of an unknown form (unspecified whether turtle or not) under "*Trionyx* sp. Owen 1842" but without any reference to *Saurodesmus robertsoni*. Attribution of *Saurodesmus robertsoni* to the Triassic turtle family Proterochersidae was first suggested by Nopcsa [32], and later in the second English translation of Zittel's "Grundzüge der Paläontologie" [33] (interpreting the bone as a femur) and by Romer [34] (interestingly, the taxon was later omitted by Romer [35]). In all these cases there was justification even though all three works treated proterochersids as amphichelydians ("primitive turtles"; e.g. [36]) not belonging to the Pleurodira (in contrast to initial interpretations; e.g. [15]). A similar approach also led Bergounioux [37] and Huene [38] to declare *Saurodesmus robertsoni* as a possible femur of a proterochersid turtle (this time classifying Proterochersidae as a family of the Pleurodira). Later, the specimen was listed as a marine primitive turtle by Swinnerton ([39] and subsequent reprints), as a possible turtle by Kuhn [40], as a right crocodilian humerus by Ingles and Sawyer [41], and as a crocodile by Benton and Walker [42]. Taylor and Cruickshank [43], based on their personal communication with G. W. Storrs, and Storrs [44] dubbed it a reptilian humerus of unknown affinity but not crocodilian. Its turtle relationship was recently questioned by Joyce [45] and this conclusion was upheld by Szczygielski [16] but no serious effort was taken to revise the specimen in recent decades. Adding to the complicated systematic history of that troublesome taxon, "*Saurodesmus*" is also an occasional misspelling of *Staurodesmus* Teiling, 1948 [46]–a genus of charophyte algae [47–51]. The aim of this work is to redescribe *Saurodesmus robertsoni*, provide new documentation of the only known specimen, and identify it taking into account new discoveries of similar patterned long bones in other amniotes made since its description.

## Institutional abbreviations

BP (ESI), Evolutionary Studies Institute (formerly Bernard Price Institute), University of the Witwatersrand, Johannesburg, South Africa; CXPM, Chuxiong Prefectural Museum, Chuxiong, China; FMNH CUP, Catholic University of Peking collection of the Field Museum of Natural History, Chicago, USA; IVPP, Institute of Vertebrate Paleontology and Paleoanthropology, Chinese Academy of Sciences, Beijing, China; MB, Museum für Naturkunde, Berlin, Germany; MCZ, Museum of Comparative Zoology, Harvard University, Cambridge, USA; NHMUK, The Natural History Museum, London, United Kingdom; NMQR, National Museum, Bloemfontein, South Africa; PIN, Palaeontological Institute, Russian Academy of Sciences, Moscow, Russia; PULR, Universidad Nacional de La Rioja, La Rioja, Argentina; SMNS, Staatliches Museum für Naturkunde Stuttgart, Stuttgart, Germany; UF H, Florida Museum of Natural History, Division of Herpetology, Gainesville, USA; UFRGS, Departamento de Paleontologia e Estratigrafia, Instituto de Geociências, Universidade Federal do Rio Grande do Sul, Porto Alegre, Brazil; ZPAL, Institute of Paleobiology, Polish Academy of Sciences, Warsaw, Poland.

## Material and methods

NHMUK PV OR 28877 comes from a glacial estuarine shale erratic of Rhaetian (Late Triassic) age from Linksfield, Elgin [1, 10, 42, 44]. It belonged to the private collection of Alexander Robertson from Inverugie, Aberdeenshire, Scotland [1, 4, 5] and was purchased by the Natural History Museum, UK, London, in 1854 [1].

The locality of Linksfield first identified by Gordon [52] was initially considered Early Jurassic (Lias) [4, 52], later an equivalent to the Late Jurassic–Early Cretaceous Purbeck or Wealden strata [2, 3, 5, 6], and eventually dated to Rhaetian [8, 53–55]. Geological studies in the locality were performed by Brickenden [56] and details summarized by Judd [10]. Beside *Saurodesmus robertsoni*, the locality has also yielded the remains of plants, invertebrates, plesiosaurs, and fish [1, 4–6, 10, 18, 42–44, 57–59].

The oldest known illustration of the specimen by Jonathan Stiven (Fig 1) states that it was found by Alexander Robertson and includes a note in pencil: "Malcolmson (Wealden Elgin) 1838 [or 1839 –not entirely clear] Proc Vol 3 p. 141". This seems to be a mix-up of two references: a report on supposed Wealden strata in Linksfield published by Malcolmson in 1838 (Proceedings of the Geological Society of London, volume 2, pp. 667–669) [2] and an abstract of his presentation from 1839 (Proceedings of the Geological Society of London, volume 3, pp. 141–144) [3] concerned with Palaeozoic localities from the area and only mentioning "the Purbeck beds at Linksfield" in passing. However, neither of these papers mentions NHMUK PV OR 28877, at least not in a clear way. Malcolmson [2] declared that "the Rev. G. Gordon has found a Saurian bone" in Linksfield, but that does not fit the caption on the illustration attributing the find to Robertson. Note that no other sources explicitly state who found the bone (Owen [4], Duff [5], and Seeley [1] only wrote that it had been in the possession of Robertson, not that he was the original discoverer) nor when it was found (Stiven's illustration is captioned "Elgin 1840" but the layout of the text suggests that this is the date when the watercolour was produced, not necessarily the date of the finding). Therefore, there is a slight chance that this is the same bone that was indeed found by Gordon and, e.g., given to Robertson, that they found it together, or that the attribution of the find to either Gordon (by Malcolmson [2]) or Robertson (by Stiven) was in error. Alternatively, this may be a separate specimen–in such a case, however, Gordon's specimen would either have to be a plesiosaur neural arch figured by Duff [5] (pl. V, Fig 2; no owner named) or it probably never resurfaced in the scientific literature; although the Linksfield strata contain plesiosaurs, the only other

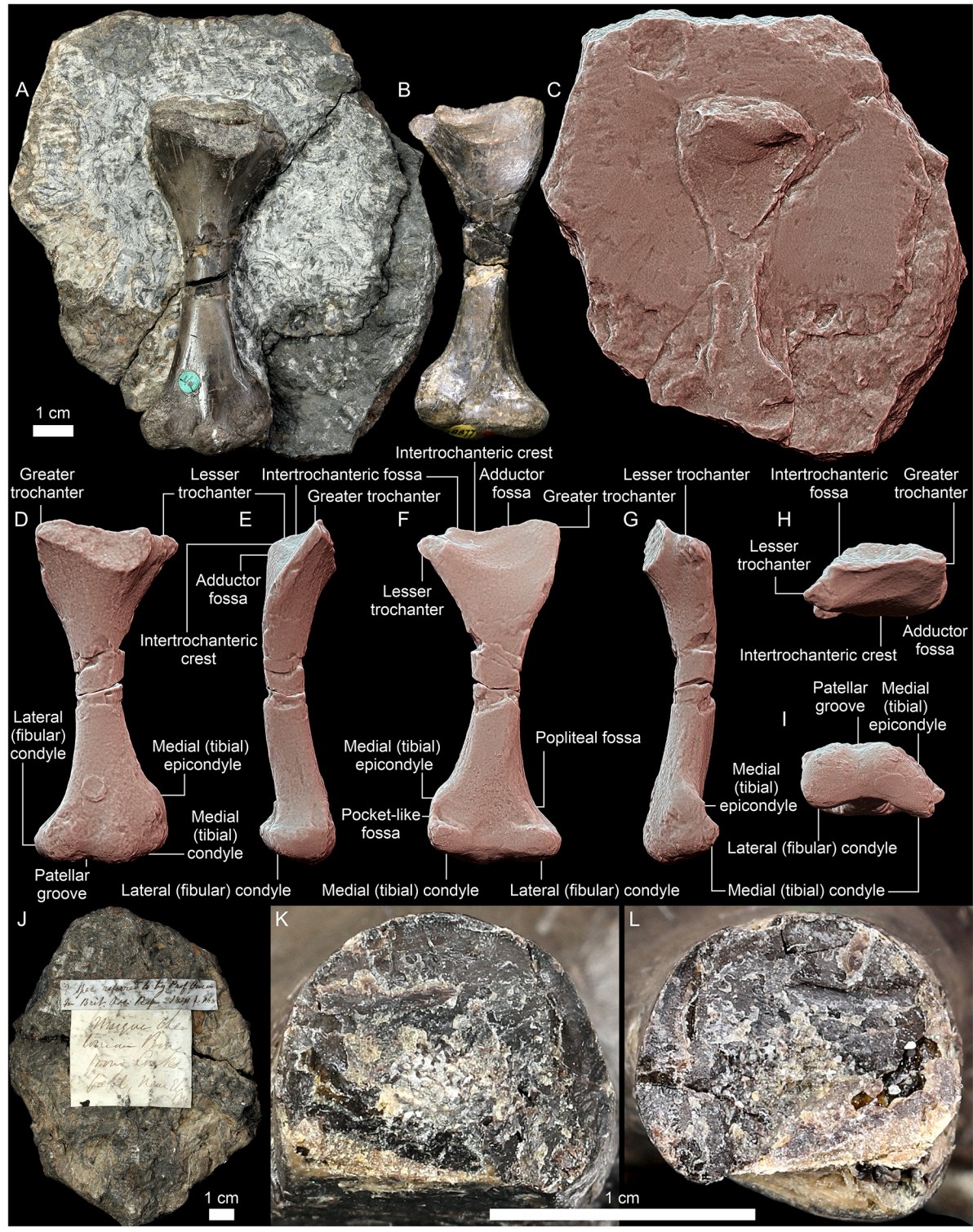

**Fig 2.** *Saurodersmus robertsoni* NHMUK PV OR 28877, right femur (A–B, D–I, K, L) in anterior (A, D), posterior (B, F), lateral (E), medial (G), proximal (H), and distal (I) view, closeup of the natural cross-section through the shaft (K, L), and associated rock fragment with natural mould of the posterior surface of the bone (A, C, J) with the bone in place (A) and removed (C), and backside (J). Two historical labels on the rock read "? Spec. referred to by Prof. Owen in Brit. Ass. Rep. 1841 p. 168" and "Unique Chelonian Bone from Linksfield, near Elgin" (J) Panels C–I are presented with the Radiance Scaling (Lit Sphere) shader enabled to enhance the geometric detail.

published bone specimens were vertebrae found by John Martin and Charles Moore [5, 43, 54, 57]. Publication of the full version of Malcolmson's [3] communication was postponed, as it was decided by the Council of the Geological Society that fish material should be first studied and described by Louis Agassiz, and its final rendition (still mentioning Linksfield only in passing and not referring to NHMUK PV OR 28877) was assembled fifteen years after Malcolmson's death from recovered manuscript and geological sections [60]. If the intended use of Stiven's watercolour of NHMUK PV OR 28877 was to accompany Malcolmson's work in any way, it was never fulfilled.

NHMUK PV OR 28877 and its associated natural mould as well as the humeri of *Stagonolepis olenkae* Sulej, 2010 [61] ZPAL Ab. III/1175 and *Parasuchus* sp. ZPAL Ab. III/1994, femora of *Bienotherium yunnanense* Young, 1940 [62] FMNH CUP 2285, *Bradysaurus baini* (Seeley, 1892) [21] NHMUK PV R 4066, *Cynognathus crateronotus* NMQR 1206, Cynognathia indet. (*Cynognathus* sp. or *Diademodon* sp.) BP/1/1675, *Langbergia modisei* Abdala, Neveling, & Welman, 2006 [63] NMQR 3255, *Moschorhinus kitchingi* Broom, 1920 [64] NMGR 3939, and *Tritylodon longaevus* Owen, 1884 [65] BP/1/5089 and BP/1/4783, and humerus and femur of *Palaeochersis talampayensis* Rougier, Fuente, & Arcucci, 1995 [66] PULR 068, were digitized using photogrammetry by photographing each piece and generating sparse clouds, depth maps, and mesh in Agisoft Metashape 1.8.4.14856 Pro. Humeri and femora of *Proterochersis porebensis* Szczygielski & Sulej, 2016 [67] ZPAL V. 39/50, ZPAL V. 39/156, and ZPAL V.39/432, *Proganochelys quenstedtii* Baur, 1887 [68] SMNS 16980 and SMNS 17203, *Labidosaurus hamatus* (Cope, 1895) [69] MB.R.4433, and *Silesaurus opolensis* Dzik, 2003 [70] ZPAL Ab. III/452 were digitized using the Shining 3D EinScan Pro 2X 3D scanner fixed on a tripod with Ein-Turntable (alignment based on features), and EXScan Pro 3.2.0.2–3.7.0.3 software. The number of turntable steps was varied, chosen depending on the specimen. 3D models of the humerus and femur of *Trionyx triunguis* (Forskål, 1775) [71] UF H65522 (UF:Herp:65522; ark:/87602/m4/454288 and ark:/87602/m4/454285), femur of *Chelus fimbriata* Schneider, 1783 [72] UF H85199 (UF:Herp:85199; ark:/87602/m4/450855), and humerus and femur of *Sphenodon punctatus* (Gray, 1942) [73] UF H14110 (UF:Herp:14110; ark:/87602/m4/M11010) were generated based on CT-scans performed on General Electric phoenix v|tome|x m 240 in the UF Nanoscale Research Facility (University of Florida), available on MorhoSource (projects "Macroevolutionary patterns of shape evolution in turtles" and "Digitizing the Florida Museum of Natural History's Herpetology collections", NKC 1.0). The 3D model of the humerus of *Crocodylus porosus* Schneider, 1801 [74] (Laboratory of Stephen Wroe, l-sw:xcb: Cp4; ark:/87602/m4/M114012) was generated based on CT-scans performed on Siemens Somatom Definition AS+ in I-MED Radiology (I-MED Radiology Armidale) and is available on MorhoSource [75] (CC BY 4.0). For figures, snapshots of the 3D models were captured in MeshLab 2021.10 [76] in orthographic view with Radiance Scaling (Lambertian and Lit Sphere) shader [77] enabled to enhance the surface detail and lighting.

Because *Saurodesmus robertsoni* for over a century remained an enigmatic taxon and its nature eluded numerous researchers specializing in various systematic groups, its identification should not be considered trivial. For that reason, we considered a wide range of taxa. Only the most relevant from the historical or anatomical standpoint, are presented in the main text, whereas those which were never officially proposed as possible identifications of NHMUK PV OR 28877 but could be represented in Triassic faunas (younginiforms, sauropterygians, choristoderans, basal panarchosaurs, 'protorosaurs', allokotosaurs, stem archosaurs, kuehneosaurids) are discussed in the S1 File.

Comparative descriptions encompassing representatives of numerous amniote groups, frequently differing in stance and degree of torsion of stylopodial bones, especially when dealing with problematic specimens of uncertain affinity, unavoidably leads to some problems

concerning directional terminology as homologous structures and surfaces may be, e.g., variably directed posteriorly or ventrally, medially or anteriorly, etc. To avoid confusion, we prioritize nomenclature typically employed for each group and in the cited references. That means that the specimen in question needs to be considered accordingly. The terminology for cynodont femora follows the convention employed by Jenkins [78] and Guignard *et al*. [79]. Comparisons with cynodont taxa are based on personal observation of the specimens, photographs, 3D models, and previously published information [27, 78–118] (see Table 1).

No permits were required for the described study, which complied with all relevant regulations.

## Results

### Systematic palaeontology

SYNAPSIDA Osborn, 1903 [145]

THERAPSIDA Broom, 1905 [146]

CYNODONTIA Owen, 1861 [147]

?TRITYLODONTIDAE Cope, 1884 [148]

*Saurodesmus robertsoni* Seeley, 1891 [1]

Holotype: NHMUK PV OR 28877, almost complete right femur (Fig 2).

Diagnosis: NHMUK PV OR 28877 can be distinguished from all other non-mammaliaform and early mammaliaform cynodonts by the following combination of femoral features: comparatively short diaphysis, lateromedially expanded proximal and distal regions, well-developed trochanters, proximodistally long and laterally flaring greater trochanter, lesser trochanter mostly medially projected and proximally placed but well-separated from the femoral head, intertrochanteric and adductor fossa separated by a poorly-developed intertrochanteric crest, well-developed medial epicondyle, medial (tibial) and lateral (fibular) condyles similar sized in ventral aspect, a distinct dorsal eminence proximal to the medial (tibial) condyle located close to the level of the long axis of the femoral shaft and almost in the middle of the width of the distal expansion, pocket-like fossa proximal to the medial (tibial) condyle ventrally, "semicolon-shaped" distal articular surface.

Provenance and age: The holotype comes from the Linksfield locality (Elgin, UK). The bearing levels were interpreted to be of Rhaetian (Late Triassic) age [8, 53–55].

### Description

NHMUK PV OR 28877 (Fig 2) is interpreted here as a cynodont femur based on its general form, lateromedially expanded and dorsoventrally flattened proximal and distal ends with a semi-circular proximal fossa and well-developed distal condyles (ruling out non-stylopodial bones), and the absence of structures identifiable as ectepicondylar and/or entepicondylar foramen or groove (ruling out humeri of most synapsids and reptiles; see Discussion).

It is a stout element, with a relatively short and nearly straight diaphysis in dorsal/ventral view. The shaft shows some damage, particularly medially and dorsally, and has no distinct morphological features. The ventral and the distal half of the lateral surface of the diaphysis are flat whereas the dorsal surface is rounded. Its narrowest point (both in the dorsoventral and mediolateral aspects) is located around the middle of the preserved bone's length, indicating that it was slightly distal to mid-length when the bone was intact (i.e., before the proximal region was broken off).

The proximal region has a dorsal curvature regarding the diaphysis as observed medially/laterally. The proximal end is triangular in the dorsoventral aspect. Its medial margin (Fig 2G and 2H) is straight (roughly parallel to the long axis of the bone in medial view and directed

**Table 1. Cynodont taxa allowing morphological comparisons with NHMUK PV OR 28877 and relevant literature references and original specimens examined personally, as 3D models, or in previously unpublished photographs.**

| Clade | Taxon | References |
|---|---|---|
| early cynodonts | *Bolotridon frerensis* (Seeley, 1895) [27] | [27] |
| | *Galesaurus planiceps* Owen, 1860 [119] | [100, 111, 114, 115] |
| | *Procynosuchus delaharpeae* Broom, 1937 [120] | NHMUK PV R 37054, [116] |
| | *Thrinaxodon liorhinus* Seeley, 1894 [121] | BP/1/7199, [78] |
| Cynognathia | *Andescynodon mendozensis* Bonaparte, 1969 [122] | [117, 118] |
| | *Boreogomphodon jeffersoni* Sues & Olsen, 1990 [123] | [80, 118] |
| | *Cricodon metabolus* Crompton, 1955 [81] | [81, 115] |
| | *Cynognathus crateronotus* Seeley, 1895 [27] | NMQR 1206, [27, 113] |
| | *Diademodon tetragonus* Seeley, 1894 [124] | [78, 82, 83] |
| | *Exaeretodon argentinus* (Cabrera, 1943) [125] | [84, 118] |
| | *Langbergia modisei* Abdala, Neveling, & Welman, 2006 [63] | NMQR 3255 |
| | *Luangwa drysdalli* Brink, 1963 [126] | [85, 118] |
| | *Massetognathus ochagaviae* Barberena, 1981 [127] | [86] |
| | *Massetognathus pascuali* Romer, 1967 [128] | [87, 118] |
| | *Pascualgnathus polanskii* Bonaparte, 1966 [129] | [118] |
| | *Scalenodon angustifrons* (Parrington, 1946) [130] | [115, 118] |
| | *Traversodon stahleckeri* Huene, 1936 [88] | [88, 118] |
| | *Menadon besairiei* Flynn et al., 2000 [131] | [89] |
| | *Santacruzodon hopsoni* Abdala & Ribeiro, 2003 [132] | [89] |
| Probainognathia | *Aleodon cromptoni* Martinelli et al., 2017 [133] | UFRGS PV-1046-T, [90] |
| | *Brasilodon quadrangularis* Bonaparte et al., 2003 [134] | UFRGS PV-1043-T, [79, 91] |
| | *Chiniquodon omaruruensis* Mocke, Gaetano, & Abdala, 2020 [92] | [92] |
| | *Chiniquodon theotonicus* Huene, 1936 [88] | MCZ VPRA-3616, [88, 93, 94] |
| | *Irajatherium hernandezi* Martinelli et al., 2005 [95] | [95, 96] |
| | *Probainognathus jenseni* Romer, 1970 [135] | MCZ VPRA-4017, MCZ VPRA-4019, MCZ VPRA-4064, [94] |
| | *Prozostrodon brasiliensis* (Barbarena, Bonaparte, & Teixeira, 1987) [136] | UFRGS PV-0248-T, [97] |
| | *Therioherpeton cargnini* Bonaparte & Barberena, 1975 [137] | [98] |
| | *Trucidocynodon riograndensis* Oliveira, Soares, & Schultz, 2010 [99] | UFRGS PV-1051-T, [99, 101] |
| | *Sinoconodon rigneyi* Patterson & Olson, 1961 [138] | [102] |
| Tritylodontidae | *Bienotherium yunnanense* Young, 1940 [62] | FMNH CUP 2285, FMNH CUP 2286, FMNH CUP 2295, FMNH CUP 2298, FMNH CUP 2300, [103, 115] |
| | *Bienotheroides* sp. | IVPP V 7906 [104] |
| | *Oligokyphus major* Kühne, 1956 [105] | NHMUK PV R 7465, NHMUK PV R 7466, NHMUK PV R 7467, [105, 115] |
| | *Tritylodon longaevus* Owen, 1884 [65] | BP/1/4567, BP/1/4783, BP/1/5089, BP/1/5152a, BP/1/5305, BP/1/5516, BP/1/5671, [106] |
| | Tritylodontidae indet. CXPM C2019 2A235 | [107] |
| | Tritylodontidae indet.(?*Kayentatherium wellesi* Kermack, 1982 [139]) | MCZ VPRA-8838, MCZ VPRA-8839, [108] |
| Mammaliaformes | *Eozostrodon parvus* Parrington, 1941 [140] | [102, 109] |
| | *Erythrotherium parringtoni* Crompton, 1964 [141] | [109] |
| | *Haldanodon exspectatus* Kühne & Krusat, 1972 [142] | [110] |
| | *Megazostrodon rudnerae* Crompton & Jenkins, 1968 [143] | [109] |
| | *Morganucodon watsoni* Kühne, 1949 [144] | [112] |
| | Morganucodontidae indet. PIN 4774/1 | [112] |

proximomedially but without any distinct curvature) and has a rounded-off edge. The lateral margin is sharper and gently convex laterally and proximoventraly (Fig 2E and 2H). The femoral head is broken off at the base alongside the top of the greater trochanter (Fig 2A, 2E, 2G and 2H). Therefore, the shape, size, and precise direction of the head, as well as the presence and morphology of the neck are unknown. Likewise, the apex of the lesser trochanter is damaged, however, it is clearly separated from the femoral head by a very small saddle-shaped notch on the edge of the intertrochanteric fossa (Fig 2A, 2D and 2F), bearing a low but sharp ridge. The ventromedially oriented lesser trochanter is crest-like with a somewhat expanded proximal tip. The tip of the lesser trochanter was distal to the proximal tip of the greater trochanter (Fig 2F). Ventrally, the shallow intertrochanteric fossa proximally and the adductor fossa more distally are separated by a low, delicately bowed, lateroproximodorsally directed intertrochanteric crest (Fig 2F), which is more distinct medially than laterally. The adductor fossa is rounded. The dorsal surface of the femur, distal to the missing femoral head is convex, bears several shallow, longitudinal vascular grooves. The proximoventrally facing surface distal to the adductor fossa is nearly flat. There is no third trochanter.

The distal end is nearly as lateromedially wide as the proximal end and similarly flared (Fig 2). Its lateral face is flat dorsoventrally and gently concave proximolaterally proximodistally. Medially, the distal region of the bone is narrower than laterally, with a rough proximal portion. A triangular medial epicondyle with sharp edges points ventrally. The dorsal surface of the distal end of the bone projects three rounded eminences separated by two shallow, wide concavities (the lateral one being the laterally shifted patellar groove). The lateral eminence is associated with the lateral (fibular) condyle; the middle one, located at the level of the long axis of the bone, is associated with the lateral limit of the medial (tibial) condyle; and the medial one (less profound) associated with the medial epicondyle. The surfaces of the lateral (fibular) and medial (tibial) condyles and of the medial epicondyle are abraded, but the shape of the distal end of the bone is not substantially altered. The distal articular surface of the bone can be roughly characterized as "semicolon-shaped": the lateral (fibular) condyle is large and oval; the intercondylar fossa is distinct but not very deep; and the medial (tibial) condyle is elongated-ovoid, rounded dorsolaterally and pinched ventromedially, projecting further posteriorly than the lateral condyle (Fig 2I). The distal reach of both condyles is roughly the same (Fig 2A, 2B, 2D and 2F). Ventrally, the proximal extent of the articular surfaces of both condyles is comparable and relatively small, they are proximodistally shorter than they are wide (Fig 2B and 2F). The lateromedial extension of the medial (tibial) condyle is smaller than that of the lateral (fibular) one, and both reach the respective edges of the epiphysis (Fig 2F); the medial (tibial) condyle extends medially slightly past the epicondyle. The proximal edges of the condyles are well marked and associated with two small but distinct fossae (Fig 2B and 2F). Particularly the medial fossa is clearly delineated distally and medially by a fold-like ridge, which continues towards the tibial epicondyle. The popliteal fossa is very shallow and indistinct (Fig 2F).

Overall, the current state of the specimen is virtually the same as in the 1800s, except of a small crack (not shown in any historical figure) lateral to the hole in the middle of the shaft and an additional transverse break through the shaft (separate from the two old glue-filled transverse cracks shown in previous figures) which was not indicated by Stiven (Fig 1) and Duff [5] but was shown by Seeley [1], and thus possibly occurred during the process or after the bone was separated from the matrix in 1889 [1]. At least during Stiven's time (Fig 1), the hole in the middle of the shaft was not yet filled with yellow glue; Duff's [5] and Seeley's [1] figures are black and white and somewhat idealized, so provide no information in that regard.

Interestingly, Stiven's illustration (Fig 1) shows the specimen upside-down, suggesting that at least initially the notch between the femoral head and the lesser trochanter could have been interpreted as an ectepicondylar foramen, and the worn surface of the distal end as a poorly

preserved humeral head and the greater and lesser trochanters. Admittedly, until the bone was fully removed from the matrix to reveal its ventral structures, its morphology could have been even more confusing.

## Discussion

Although the taxonomic validity of *Saurodesmus robertsoni* was never questioned (despite R. Lydekker in his comment to Seeley's [1] paper, considering it "not worthy of being made a type of a genus"), its systematic placement was always troublesome. Since its discovery, NHMUK PV OR 28877 was referred or compared to representatives of a wide spectrum of amniote groups (turtles, crocodiles, generalized crurotarsans, procolophonians, mammals, dinocephalians, and non-mammalian cynodonts; Fig 3), but these attributions rarely stood the test of time and changed repeatedly. Despite these numerous attempts at identification, most recently, the taxon was deemed an indeterminate reptile [44, 45]. This uncertainty resulted not only from the incompleteness and odd morphology of the specimen, but also immense developments in vertebrate palaeontology which took place since the early 1840s, when the specimen was first studied, most notably in the state of knowledge concerning the temporal distribution, relationships, and morphological evolution of various amniote lineages. For example, at the time it was still a relatively common practice to assign Mesozoic or even Palaeozoic vertebrate finds to extant genera [4, 149–152], which shows how different the perception was of possible taxon duration and inclusiveness. Adding the much smaller array of comparative material available back then, many of the interpretations were based on studies of much younger material, frequently not representative of Triassic taxonomic and anatomical diversity, and thus at least partly speculative [1, 153, 154]. For that reason, historical arguments concerning the identity of *Saurodesmus robertsoni* need to be re-evaluated.

### Pantestudinata

As correctly noted by Seeley [1], the similarity of NHMUK PV OR 28877 to turtle limb bones are mostly restricted to the presence of a U-shaped fossa ventrally on the proximal end and a vague resemblance of the overall form. However, upon closer examination, these features strike as only superficial, and more differences become apparent. Seeley [1] listed a number of characters distinguishing NHMUK PV OR 28877 from turtle stylopodial bones (given the nomenclature used by him, mostly humeri, but most characters are also applicable to femora): (1) presence of a medullary cavity (contested by Lydekker in his comment published alongside Seeley's [1] paper); (2) straight rather than dorsoventrally (anteroposteriorly, in cynodont terms) sinuous profile; (3) supposed lateral and medial processes ("radial and ulnar crests", i.e., greater and lesser trochanters, according to our interpretation) separated by a wide fossa; (4) distinctiveness and small size of the supposed lateral process; (5) unusual development, convexity, and roughness of the supposed medial process; (6) larger thickness of the shaft at the contact with the head; (7) form of transverse curved ridges crossing the supposed intertubercular fossa; (8) incongruent morphology of the distal end: unusual broadness, flattening and concavity of the lateral ("ulnar") edge, compression of the medial ("radial") side into a convex edge, dissimilar distal articular surface, and absence of ectepicondylar groove or foramen. Nonetheless, the interpretation of NHMUK PV OR 28877 as a turtle femur gained much traction throughout the specimen's history [4–6, 8, 10, 18, 30, 32–34, 37–40], although it was almost never actually supported by morphological arguments. Only Nopcsa [30] argued for its identity as a pleurodiran femur and justified that, noting the structural resemblance of its proximal end to both the turtle humeri and femora and the lack of the ectepicondylar foramen (refuting identification as a humerus). He interpreted the more elaborate development of its

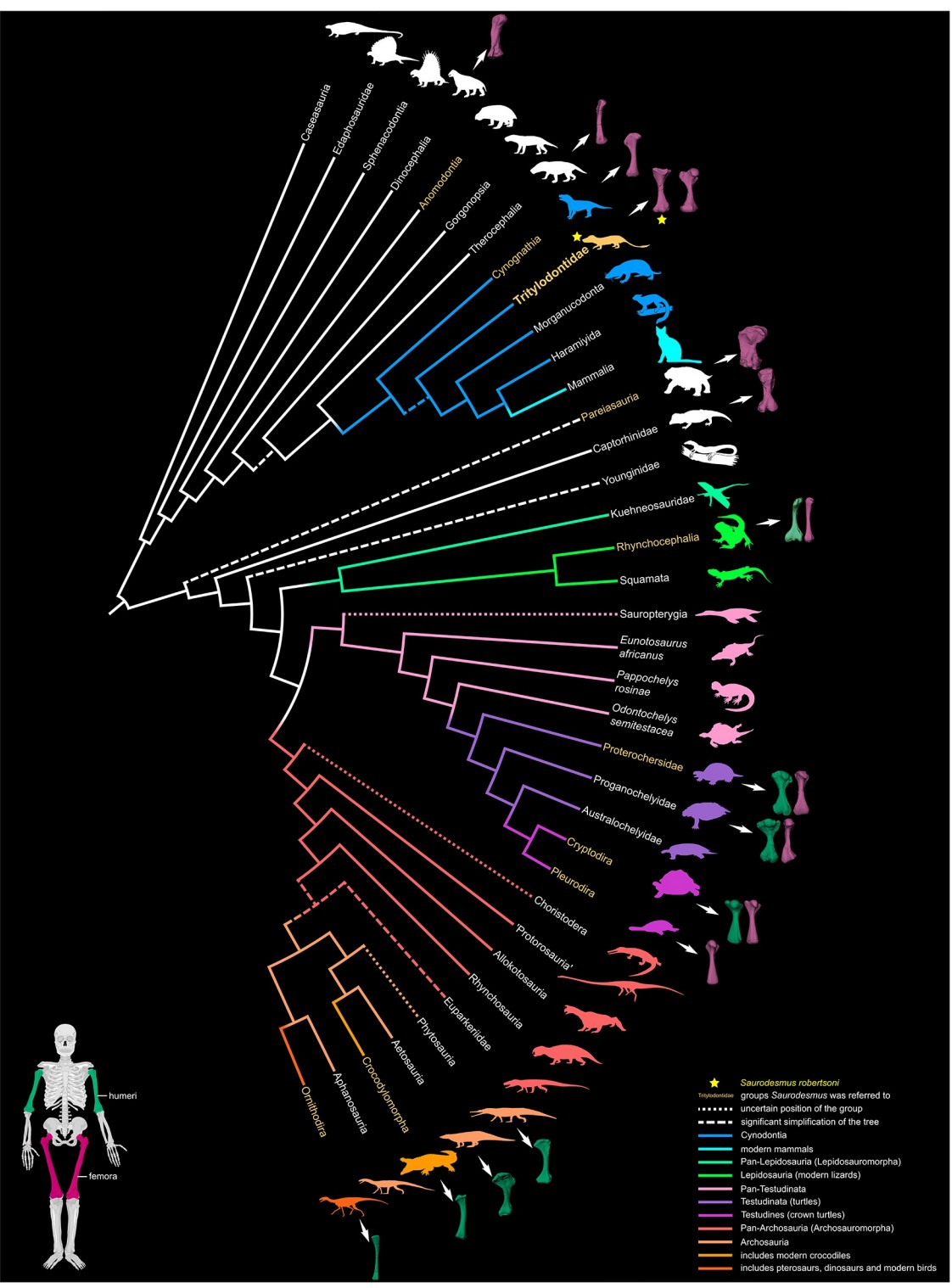

**Fig 3. Simplified phylogenetic tree of Amniota showing taxa discussed in the text, including groups that *Saurodesmus robertsoni* was referred to in the past.** Composite topology based on various sources, including [155–159]. See Acknowledgements for the sources of images.

distal articular condyles as an ancestral trait due to its Triassic age and the secondary simplification of the knee morphology in pleurodires related to their aquatic ecology. It must be stressed that Seeley's [1] assessment was based only on extant turtles (limb material of Triassic pantestudinates being completely unknown at the time), and Nopcsa [30] had at his disposal only information on the proximal end of the femur of *Proganochelys quenstedtii* (considered a cryptodire at the time) published by Jaekel [160], therefore they had no possibility of comparisons with actual Triassic turtles.

Quite surprisingly, Nopcsa [32], the English editors of the (posthumously published [161]) second English edition of Zittel's "Grundzüge der Paläontologie" [33], Romer [34], Bergounioux [37], and Huene [38] interpreted NHMUK PV OR 28877 as a possible proterochersid femur. Curiously, this assignment (or the specimen itself) is not present in the original German version of "Grundzüge der Paläontologie" ([162] and earlier editions) and likely was an initiative of A. S. Woodward, responsible for revisions and additions to the second English edition, and/or R. Broom, F. Nopcsa, or F. von Huene, who were acknowledged "for help with the fossil reptiles" [33]. In a sense, this view was first presented by Lydekker [13], who rather haphazardly suggested an affinity of NHMUK PV OR 28877 with the German Norian turtle '*Chelytherium obscurum*', which later turned out to be a synonym of *Proterochersis robusta* [16, 17]; however, at the time neither the name *Proterochersis* (introduced by Fraas [15]) nor the family Proterochersidae (named by Nopcsa [30]) existed. Lyddeker [13] was explicitly aware of *Proganochelys quenstedtii* (named by Baur [68]) and convinced of its taxonomic distinctiveness from '*Chelytherium obscurum*', therefore the reasoning behind this referral is unclear. Curiously, the assignment of NHMUK PV OR 28877 to the Proterochersidae was never justified in any way; in fact, realistically it never could have been, due to the complete lack of limb material attributable to the Proterochersidae at the time, which did not stop Nopcsa [32] from listing "femur with wings on proximal end" as one of the diagnostic characters of the Proterochersidae. The authors might have already been familiar with the general morphology of Triassic turtle humeri ([163]: probably *Proganochelys quenstedtii*) but their insight into femoral anatomy most likely was still limited, both taxonomically and when it comes to specimen number and quality. In addition to Jaekel's [160] partial femur of *Proganochelys quenstedtii* housed in the Berlin Museum, in 1956 (the publication date of Huene's "Paläontologie und Phylogenie der Niederen Tetrapoden" [38]) the Staatliches Museum für Naturkunde Stuttgart already had in its collection limb bones of the same species (a badly crushed distal end of femur SMNS 15759 and much better preserved and more complete bones of SMNS 16980, SMSN 17203, and SMNS 17204; [164–167]) but their first detailed description and documentation was published decades later [168] and the first stylopodial skeleton of proterochersids was discovered only very recently [67, 169, 170]. Its recovery enables testing of the hypothesis about the proterochersid identity of *Saurodesmus robertsoni*. Moreover, recent discoveries of non-testudinate pantestudinates [171–173] and new Triassic turtle species [66, 67, 174] allow a better insight into the variability and evolution of limb bones at the brink of turtle lineage and re-evaluation of the arguments for and against the turtle relationships of *Saurodersmus robertsoni*.

As correctly noted by Seeley [1] and Nopcsa [30], the absence of the ectepicondylar foramen or groove generally discredits NHMUK PV OR 28877 as a turtle humerus, since the groove or foramen is present in all known representatives of the Pantestudinata [163, 168, 171, 172, 175–177] (Fig 4) with the unique exception of *Pappochelys rosinae* Schoch & Sues, 2015 [173, 178]; even in small humeri of supposedly juvenile individuals of Triassic turtles (*Proganochelys quenstedtii* SMNS 17203 [168]; *Proterochersis porebensis* ZPAL V. 39/442, ZPAL V. 39/444, ZPAL V. 39/446 [170]), a sharply defined, deep ectepicondylar groove spans along most of the dorsal surface of the distal humeral expansion, therefore its absence in NHMUK PV OR 28877 cannot be explained by ontogeny or surface damage. *Eunotosaurus africanus* Seeley, 1892

[179] has both the ectepicondylar foramen and the entepicondylar foramen [177], the latter of which is absent in Triassic Pantestudinata. The absence of foramina or grooves, however, is not the only difference between NHMUK PV OR 28877 and pantestudinate humeri. As noted by Seeley [1], the intertubercular fossa in pantestudinate humeri lacks transverse ridges [163, 168–170, 175–178, 180, 181], NHMUK PV OR 28877 is straighter than Triassic pantestudinate humeri, and its distal end lacks the characteristic deflection opposite to the proximal end, which is observed in all Triassic and later pantestudinates [163, 168, 170–172, 175, 178] (Fig 4). Although the exact angle at which the proximal head was set in NHMUK PV OR 28877 relative to the shaft is uncertain [1], the whole proximal expansion is in fact more parallel to the distal expansion than in any pantestudinate with known humeri [168, 170–172, 175–178, 180]. Also the asymmetry of the distal end, with one edge flattened and concave proximodistally and the other convex and sharpened, as well as the wide (spanning the whole width of the bone) but restricted proximally articular surfaces of the distal condyles were correctly indicated by Seeley [1] as incongruent with pantestudinate humeri. All Triassic pantestudinates have their distal humeral expansions relatively symmetric with two roughly symmetrically developed epicondyles and straight or only gently curved and mostly rounded edges [163, 168, 170–172, 175, 176, 178]. In *Eunotosaurus africanus* and all Triassic Pantestudinata in which the ventral surface of the humeri is exposed and well ossified articular surfaces, the trochlea and capitellum are restricted in their mediolateral extent, clearly separated from the maximum mediolateral extrema of the epicondyles, and never as wide relative to their proximal extent as the articular surfaces in NHMUK PV OR 28877 [163, 168, 170, 171, 175–178]. Finally, in the humeri of Triassic Pantestudinata the medial condyle is never as pinched out as the medial condyle in NHMUK PV OR 28877 [168, 170, 175] and the epicondyles lack separate dorsal eminences, so the dorsal surface of the distal femoral end bears two rather than three eminences separated by a single rather than two fossae [168, 170, 172, 176, 175, 178]. Seeley [1] also formulated several arguments regarding the shape and size of the proximal processes and the supposed intertubercular fossa, as well as the thickness of the shaft. These characters are variable in Pantestudinata, including Triassic taxa, although in *Saurodesmus robertsoni* the proximal processes are indeed poorly developed (small, thin, and weakly protruding) compared to the lateral and medial humeral processes of Triassic turtles and whereas the fossa is not particularly wide relative to the Triassic pantestudinate intertubercular fossae, it is comparatively very shallow [163, 168–171, 175, 176, 178, 181].

Interpretation of NHMUK PV OR 28877 as a pantestudinate femur is seemingly more appealing due to the lack of foramina or grooves trab (Fig 5), but most of the other features listed above also preclude such an identification, even if they are less clearly expressed in pantestudinate femora than in humeri. The femur of *Eunotosaurus africanus* exhibits a completely different form–it is straight, more rod-like, with weaker flattening of the proximal and distal end and much less distinct morphological features [177]. In *Pappochelys rosinae*, the femur is relatively straight but slenderer and less expanded at the ends than NHMUK PV OR 28877, with more elongate, very narrow, and deeper intertrochanteric fossa [173, 178]. In *Eorhynchochelys sinensis* Li *et al.*, 2018 [172] and *Odontochelys semitestacea* Li *et al.*, 2008 [171], the curvature along the shaft is comparable as in NHMUK PV OR 28877 [171, 172], but in Testudinata it is stronger, which is particularly visible in more pronounced deflection of the distal end relative to the femoral head [67, 168, 175, 181]. Femoral torsion is less severe than humeral torsion but still higher in all Triassic pantestudinates than in NHMUK PV OR 28877 [168, 170–172, 175, 178] and most pronounced particularly in Triassic non-testudinate pantestudinates [171, 172, 178]. The morphology of the trochanters and intertrochanteric fossa is varied in Pantestudinata but generally in Triassic species the fossa is narrower, deeper, and devoid of transverse ridges, the trochanters in proximal view form a distinct angle of 90° or

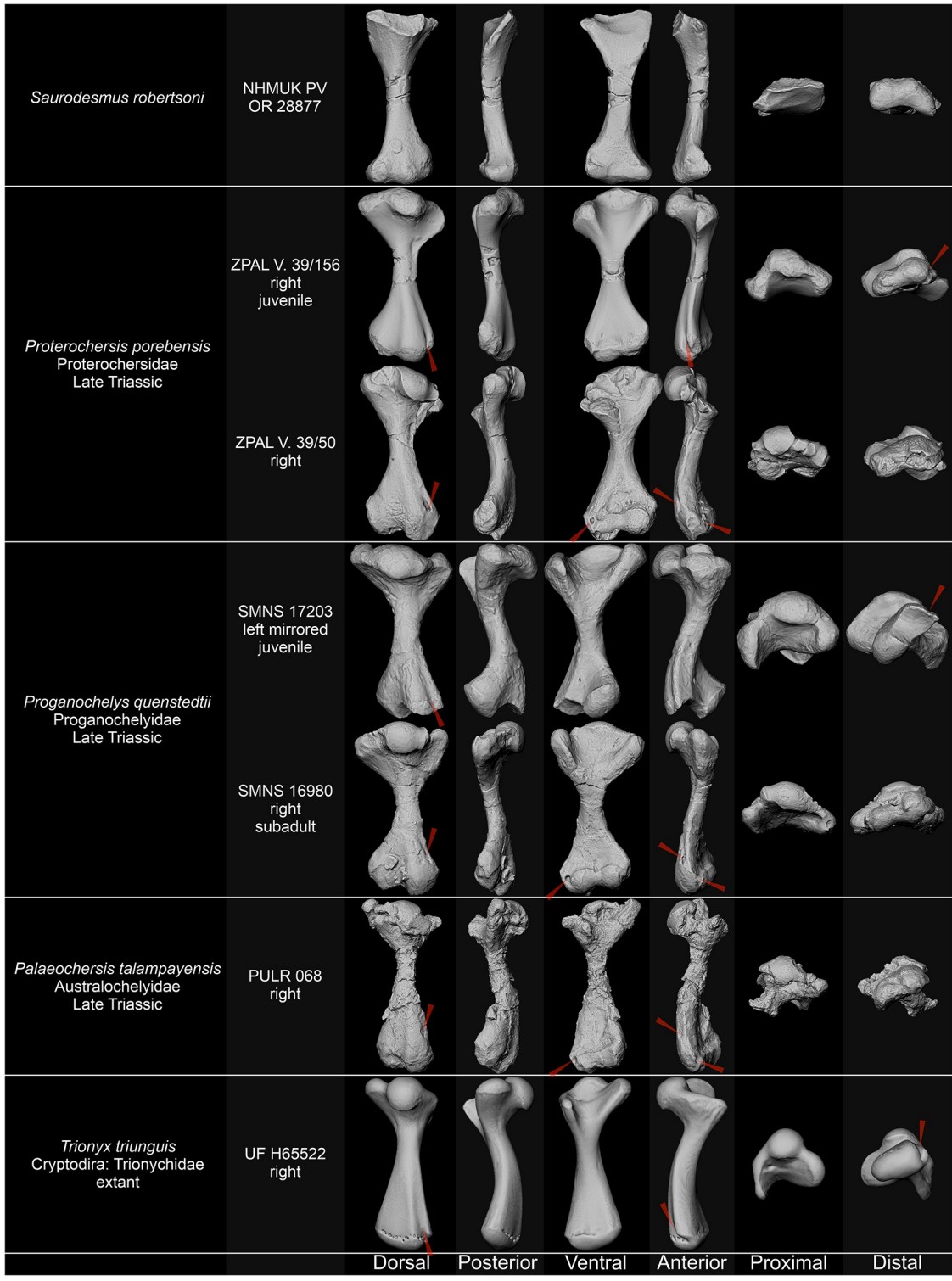

**Fig 4. Comparison of *Saurodersmus robertsoni* NHMUK PV OR 28877 with humeri of turtles.** Ectepicondylar grooves/foramina indicated by red arrowheads. Not to scale.

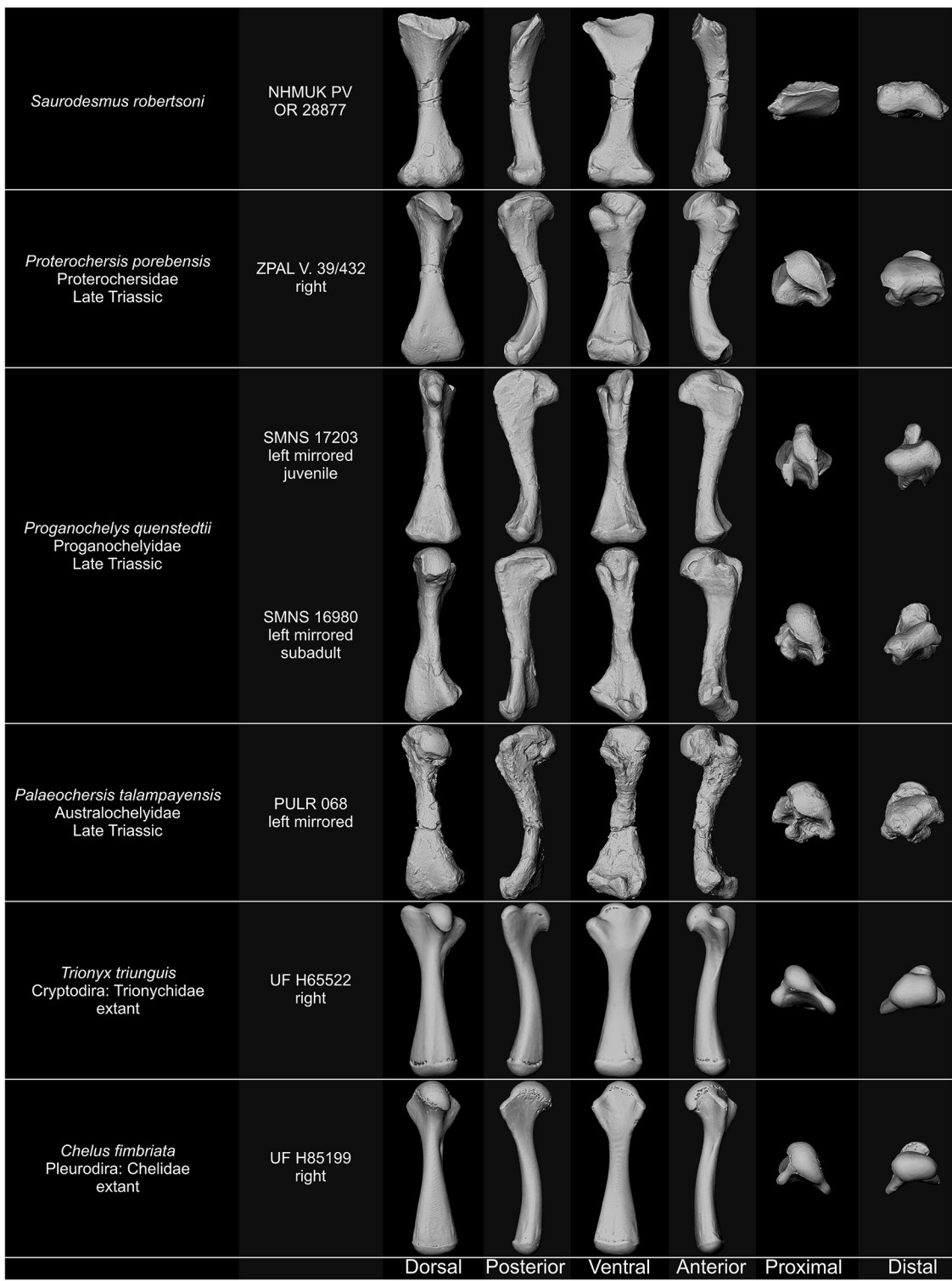

**Fig 5. Comparison of *Saurodersmus robertsoni* NHMUK PV OR 28877 with femora of turtles.** Not to scale.

less, and particularly the greater trochanter is more upright and thicker than in NHMUK PV OR 28877 [67, 160, 168, 172, 175, 178, 181, 182]. The proximal end of the Triassic pantestudinate femora is also bulkier than in NHMUK PV OR 28877 [67, 160, 168, 171, 175, 178, 181, 182]. The posterior flattening of the shaft resembles the condition seen in Triassic pantestudinate femora [67, 168, 175, 178, 181, 182] but the structure of the distal end is mediolaterally reversed relative to NHMUK PV OR 28877: in pantestudinates, the anterior (lateral) face of the distal expansion is typically flattened and may be concave whereas the posterior (medial) edge is sharpened [67, 168, 172, 175]. Furthermore, in Triassic pantestudinates the distal femoral condyles are not aligned mediolaterally with the edges of the distal expansion as in NHMUK PV OR 28877 but the lateral (fibular) condyle is located closer to the midwidth of the distal expansion and further distally than the medial (tibial) condyle [67, 168, 169, 172, 175, 178]. Unlike in NHMUK PV OR 28877, both femoral condyles in pantestudinates project medially distinct ridges along the ventral surface of the distal expansion, thus the popliteal fossa is well delineated and medially (posteriorly) a second triangular fossa is present more posteriorly, between the ridge of the lateral (fibular) condyle and the fibular epicondyle [67, 168, 169, 172, 175, 178]. Finally, there is no epicondylar eminence dorsally [67, 168, 171, 172, 175, 178]. Thus, the distal articular morphology in Triassic Pantestudinata is indeed more elaborate than in many extant taxa, as speculated by Nopcsa [30], but its configuration is different than in *Saurodesmus robertsoni*.

We were not able to unambiguously confirm the presence of a distinct medullary cavity in NHMUK PV OR 28877 –at the moment of our study the bone was separated into two parts, but the surface of the break was obscured by glue remnants and sediment particles. However, the middle region of the shaft appears to be filled with thin trabeculae (Fig 2K and 2L). Regardless, the differences in morphology extending beyond the variability observed within the Pantestudinata and not consistent with observed evolutionary trajectories between the earliest known pantestudinates (*Eunotosaurus africanus*) and Triassic Testudinata allow to exclude the possibility that *Saurodesmus robertsoni* is a representative of that lineage.

## Parareptilia

Parareptilian resemblance of *Saurodesmus robertsoni* have been indicated twice in the literature. Seeley [21] perceived NHMUK PV OR 28877 to be similar to the femur of *Bradysaurus* ('*Pareiasaurus*') *baini* (Fig 6) based on the shape of the trochanters and the presence of a rugosity on the laterodorsal surface, which led him to the idea that NHMUK PV OR 28877 may represent an anomodont femur (note that he considered the Anomodontia to include at least the theriodonts, dicynodonts, placodonts, pareiasaurs, procolophonids, and mesosaurs). Seeley [183] also noted some resemblance of the intertrochanteric fossa on the proximal end of NHMUK PV OR 28877 with that on the femur of *Procolophon* sp. (also comparing it to the homologous structure of birds, mammals, and turtles), although he earlier considered procolophonid humeri as anatomically different [1] from NHMUK PV OR 28877.

Incidentally, the characters pointed out by Seeley as common between the procolophonids and pareiasaurs and *Saurodesmus robertsoni* are among the characters historically proposed as suggesting phylogenetic relationships between all of those taxa and turtles [184–187]. They seem to simply reflect a shared similarity resulting from various degrees of modification of an ancestral amniote morphology and perhaps some biomechanical locomotor parallelisms rather than strongly indicating a shared ancestry, considering the currently accepted phylogenies [188, 189].

Despite having those basic anatomical components and their significant dorsoventral compression, pareiasaur femora differ from NHMUK PV OR 28877 in a number of ways: they are

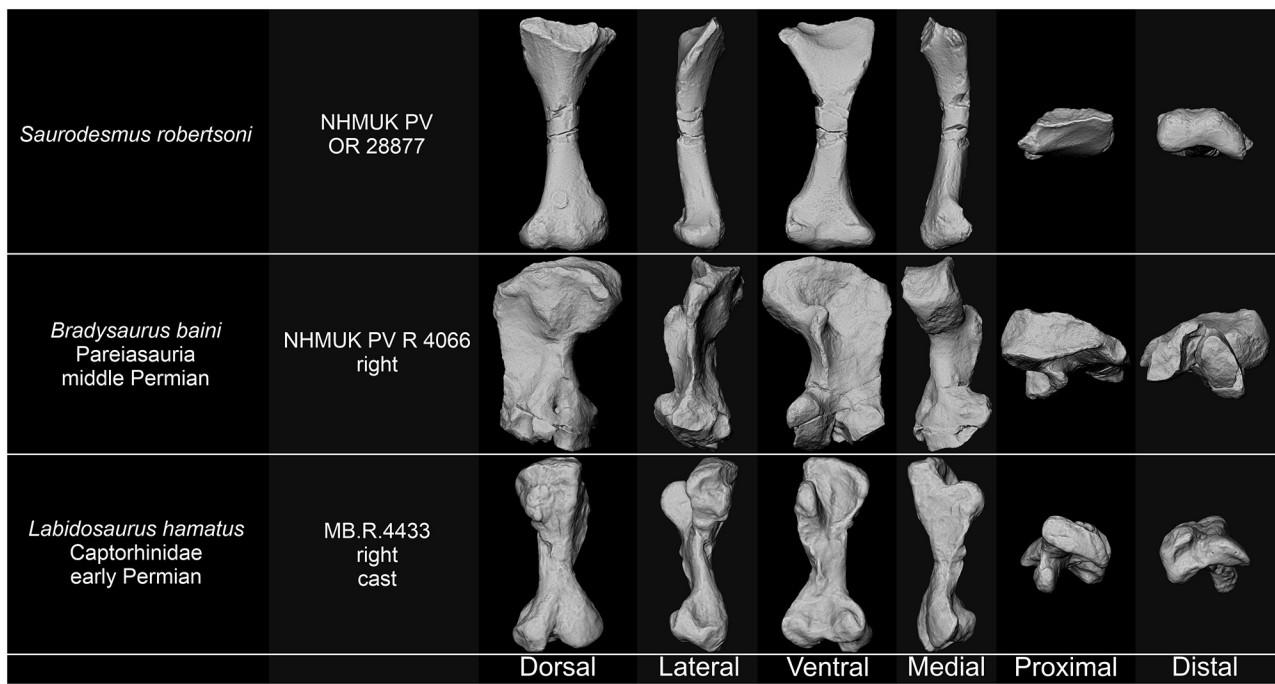

**Fig 6. Comparison of *Saurodersmus robertsoni* NHMUK PV OR 28877 with femora of non-diapsid reptiles.** Not to scale.

much broader, the result of a significantly wider shaft, which is at least 50% as wide as the proximal and distal expansions; the direction of the femoral head is subparallel to the plane of the distal expansion; the internal trochanter is located more distally and is subperpendicular to the proximal expansion; the greater (major) trochanter typically extends distally along most of the shaft's length a flattened postaxial flange (usually resulting in the posterior edge being straighter than the anterior edge); the internal (lesser) trochanter and adductor ridge is usually well developed, elongated, often sigmoid-shaped and somewhat medially positioned, away from the anterior (preaxial) edge of the femur; the dorsal surface of the distal expansion is typically pierced by a large and deep nutrient foramen and groove, elongated along the direction of the shaft; the distal end is not significantly flattened; the distal lateral condyle often extends significantly beyond the distal medial condyle (related to a relatively sprawling or oblique hind-limb posture); the articular surfaces of the condyles are longer than wide, oval in outline, with sharply defined, flattened, and substantial ventral exposure, and are separated by a deep intercondylar groove (sulcus) [184, 185, 187, 190–197] (Fig 6). There is no Triassic record of pareiasaurs, since the clade perished at the Permo-Triassic boundary [190, 191, 194].

In comparison to *Saurodesmus robertsoni*, the femur of procolophonoids is more rod-like, its proximal end is more massive and less expanded, the trochanters are more parallel to the long axis of the bone, the intertrochanteric fossa is deeper and narrower, the shaft is thicker, the popliteal and intercondylar fossae are much deeper and better defined, the distal condyles are narrower, and the edges or the bone are rounded rather than sharpened [183, 198–203]. Unlike pareiasaurs, procolophonoids did persist into the Triassic [199].

### Non-diapsid Eureptilia

The humeri of earliest eureptiles, such as captorhinids or *Protorothyris archeri* Price, 1937 [204], retain numerous plesiomorphic characters, e.g., poorly differentiated humeral head,

proximal and distal expansions set at about a right angle, massively expanded, plate-like distal end, retention of the ectepicondylar foramen, and presence of a prominent, ovoid capitellum [168, 205–208], and thus are dissimilar to NHMUK PV OR 28877. Conversely, the femora (Fig 6) correspond with NHMUK PV OR 28877 in some general, plesiomorphic characters, such as the presence of a distinct intertrochanteric fossa and lesser trochanter or distinctly expanded distal end, as well as the thickness of the lateral epicondyle and sharpening of the medial epicondyle. They differ, nonetheless, e.g., in more ventrally directed lesser trochanter, less dorsally offset femoral head, narrower and thicker proximal end, larger robustness and stoutness, clearly bipartite rather than tripartite distal end with deeply incised patellar groove, and the distal projection of the lateral condyle beyond the medial condyle [168, 205, 206, 208]. There are no representatives of non-diapsid eureptiles known from the Triassic [208, 209].

## Panarchosauria

**Rhynchosauria.** Seeley [1] simply deemed rhynchosaurian humeri dissimilar to *Saurodesmus robertsoni* but did not elaborate. Since the Rhynchosauria is represented in the Late Triassic of the Elgin area [210–212], it deserves consideration. Humeral anatomy of rhynchosaurs shares with NHMUK PV OR 28877 the presence of the proximal and distal expansion but differs in the presence of a wide and continuous proximal articular surface encapsulating the medial and lateral processes, proportionally larger proximal end, large torsion of about 90˚, presence of the ectepicondylar foramen, symmetry of the distal expansion with distinct anterior and posterior fossae, and roundness of the distal condyles and entepi- and ectepicondylar edges [211, 213, 214]. In fact, NHMUK PV OR 28877 is slightly more reminiscent of rhynchosaur femora, due to their less expanded proximal end, distinct internal trochanter, lack of pronounced torsion, simpler morphology of the distal end and wider than long articular surfaces of the distal condyles with anterior condyle extending slightly more proximally than the posterior condyle; however, it is slenderer, lacks the clearly ventral direction of the internal trochanter, defined adductor crest, and triangular fossae in the middle of the ventral and dorsal surface of the distal end, and differs in the non-equidimensional distal condyles, less rectangular distal articular surface, and sharpened anterior edge of the distal end [211, 213, 214].

**Pseudosuchia.** Seeley [1] originally identified *Saurodesmus robertsoni* as a member of "a primitive crocodilian stock" representing a new suborder of Crocodylia with a mix of crocodilian and lacertilian characters. He noticed some resemblances of NHMUK PV OR 28877 with the humeri of extant gavials and crocodiles (Fig 7), as well as fossil crocodiles, of which he mentioned *Crocodilus hastingiae* Owen, 1848 [215] (an Eocene crocodile from the United Kingdom, now *Diplocynodon hantoniensis* [216]). Due to differences in used terminology, his description is confusing for the modern reader, making it difficult to recognize the specific characters he had in mind. Because he treated the bone as a humerus, the structure referred by him as the radial process/crest is probably the lesser trochanter (in his understanding, the deltopectoral crest–in respect to crocodile humerus; lateral process–in respect to the humeri of the Pantestudinata). Consequently, his ulnar border and ulnar tuberosity probably mean the posteromedial edge of the proximal part of the bone, which is identified here as the greater trochanter (in his understanding, the internal tuberosity–in respect to crocodile humerus).

Seeley [1] noted three similarities between NHMUK PV OR 28877 and crocodilian humeri, namely (1) the sharp inner prolongation of his radial process, which probably translates to the sharp proximal edge of the intertrochanteric fossa, as identified here (in his understanding, proximal surface of the deltopectoral crest in the crurotarsan humerus); (2) prolongation of the proximal articular surface beyond his radial process (i.e., the articular surface of the head located proximally to the lesser trochanter, as identified here); (3) hollow limb bones (by

which he probably mean a well-developed medullary cavity, in contrast to turtles). However, he also noted more differences, namely that: (1) his radial crest is flexed too much dorsally–the deltopectoral crest points ventrally in crocodiles and is perpendicular to the humeral head (e.g., *Diplocynodon hantoniensis* humerus NHMUK OR 30206; [217]); (2) his ulnar (i.e., lateral, according to our interpretation) border is compressed to a sharp muscular edge (internal tuberosity and proximal posteromedial border is robust and has a smooth edge in crocodiles); (3) the proximal and distal ends are more expanded than in crocodiles; (4) the bone is straight (lack of bending of the shaft–in *Diplocynodon hantoniensis* NHMUK OR 30206 the shaft of the humerus forms a delicate arch; [217]); and (5) the condyle on his radial (i.e., medial, according to our interpretation) side is compressed in comparison to crocodiles (which likely refers to the ventrally concave area and the sharpened edge of the tibial epicondyle; lateral condyle in respect to the crurotarsan humerus). Nevertheless, beside the compression of his ulnar (i.e., lateral) margin of the proximal end (considered by him an apomorphy relative to all other reptiles), he claimed the overall morphology homologous, and therefore settled for identification of *Saurodesmus robertsoni* as a member of the Crocodylia representing a new suborder with a mix of crocodilian and lacertilian characters.

There is a vast difference in morphology between the humerus and femur in crocodiles and their kin, such as phytosaurs and aetosaurs. The most significant is that the femora have a distinctly sigmoidal shaft, the femoral head is directed medially and substantially twisted in respect to the condyles, and there is the fourth trochanter (a knob-like feature, attachment area for the muscles retracting hind limbs) on the posterior surface of the shaft [217–221]. Therefore, the interpretation of NHMUK PV OR 28877 as a crocodilian femur can be rejected straight away and it was never discussed in the literature. The resemblance of NHMUK PV OR 28877 to the crocodilian humerus is also only superficial and restricted to the general form of the bone with its expanded proximal and distal ends, as well as its relatively straight shaft. However, in crocodiles and their relatives, the expansion of the ends is not as symmetrical as in NHMUK PV OR 28877, but rather more pronounced medially than laterally (Fig 7) [217–223]. Beside the perpendicular orientation of the deltopectoral crest noted as a difference form the crocodiles by Seeley [1], both the deltopectoral crest and the internal tuberosity are large structures in the latter, being areas of attachments for the muscles responsible for major movements of the forelimbs [218–222, 224, 225]. Finally, there are no structures that can be referred to the ecto- and entepicondylar grooves or foramina, however, in crocodiles these might be poorly developed [217].

Interestingly, Seeley [1] mentioned that neither of the other reptile species from the Elgin Sandstone has the same resemblance to crocodiles as NHMUK PV OR 28877, which is puzzling in the context of the aetosaur *Stagonolepis robertsoni* Agassiz, 1844 [59], an armoured member of the Pseudosuchia, the clade that includes ancestors of modern crocodiles [226, 227]. At the moment of Seeley's publication, the resemblance of *Stagonolepis robertsoni* to crocodiles has been already noted by Huxley [228–230]. However, the proximal end of the humerus in aetosaurs is much more expanded transversely compared to other pseudosuchians (Fig 7) [221, 222], which was also noted by Seeley [1], and the exact relations between various groups now included within the Pseudosuchia were not sorted at the time [226, 227], which may partly explain Seeley's [1] opinion.

It seems that Seeley's [1] identification of NHMUK PV OR 28877 as a crocodilian humerus was the only one that was supported by some discussion, and other mentions of the specimen as a crocodilian are only repeating his results [28, 29, 42, 43]. Huene [28] arbitrarily assigned *Saurodesmus robertsoni* to his "Crurtoarsi", probably following the idea that if it is a crocodilian, it should be related to other crocodile-like reptiles. At that moment it was also considered that *Stagonolepis robertsoni* was related with other crocodile-like animals from the Germanic

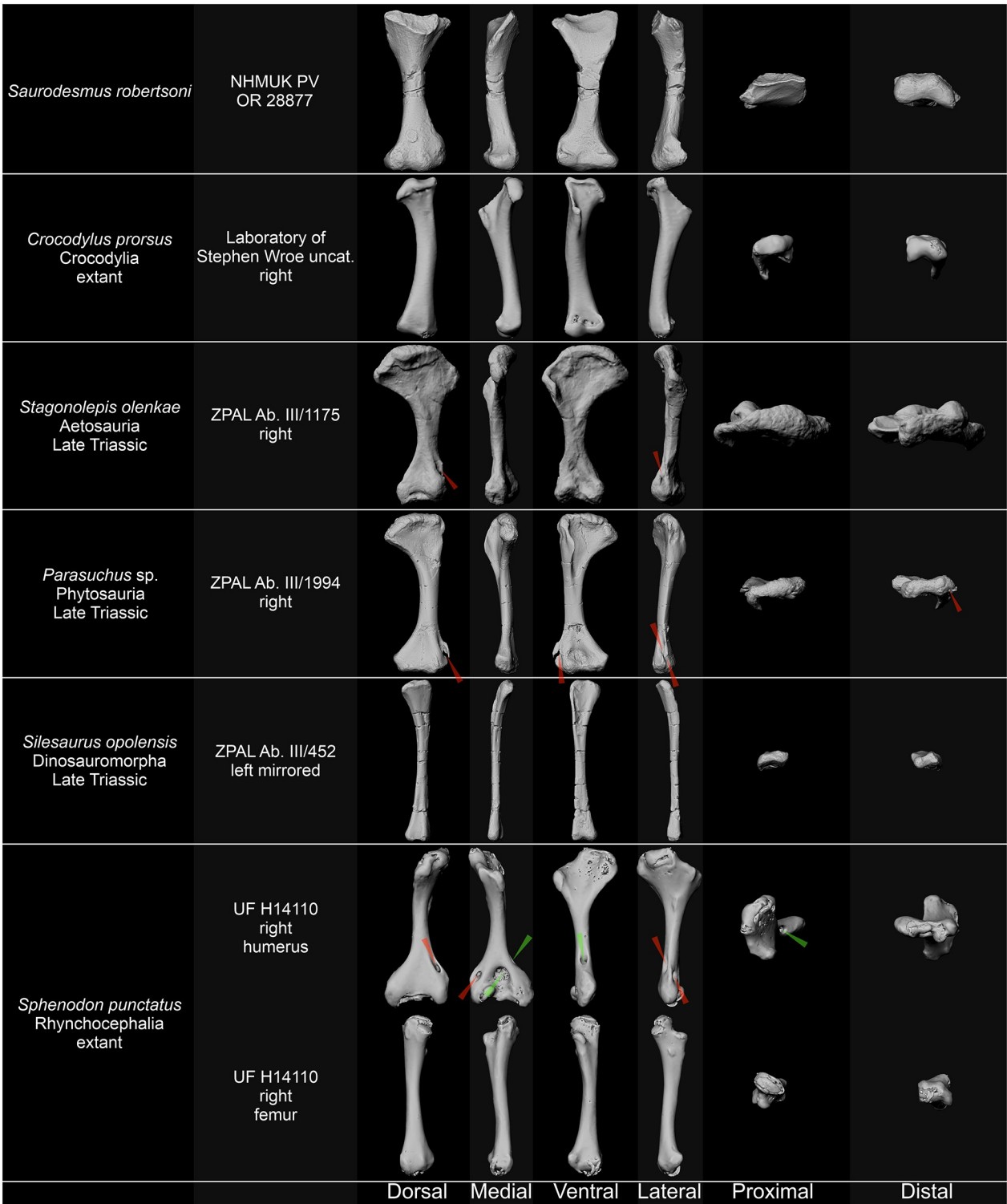

**Fig 7. Comparison of *Saurodersmus robertsoni* NHMUK PV OR 28877 with humeri of archosaurs and with humerus and femur of a rhynchocephalian.** Ectepicondylar groove or foramen indicated by red arrowheads, entepicondylar foramen indicated by green arrowheads. Not to scale.

Basin, especially the genus *Belodon* [228–230]. However, this conclusion was mostly based on the morphology of osteoderms, and the dermal armour asigned at that time to *Belodon* was later proven to be the dermal armour of an aetosaur [231–234], therefore the commonly accepted, at the time, close relationships between the various archosauromorphs were overstated.

**Ornithodira.** According to Seeley [1], NHMUK PV OR 28877 exhibits some resemblance to pterosaur humeri in characters in which it is divergent from the crocodilians. This vague statement is difficult to address, but aside of the strongly mediolaterally expanded proximal end and relative gracility, there is little meaningful similarity–NHMUK PV OR 28877 lacks the anteroposterior expansion of the proximal end, subrectangular shape of the distal articulation surface, and the trochlear form of the distal articulations which were present in early Mesozoic pterosaurs [235–239].

Lagerpetids lack the ectepicondylar foramen or groove and have much simpler humeral morphology than pterosaurs, but differ from NHMUK PV OR 28877 in only very slightly deflected proximal articular surface, medial deflection of the whole proximal end, barely expanded, more rectangular in articular view distal end, distal condyles separated by a pronounced groove, distally protruding medial condyle, greater torsion, and overall much more gracile and slender form [240–242]. Likewise, dinosauromorph humeri can be distinguished from the holotype of *Saurodesmus robertsoni* based on their overall slenderness, less deflected, predominantly proximally directed humeral head, slight medial deflection of the proximal end, lack of a separate lateral process, and barely expanded distal end with two rounded, similarly shaped and sized distal condyles (Fig 7) [70, 224, 242–245].

## Lepidosauromorpha

A lacertilian humerus identity of NHMUK PV OR 28877 was refuted by Seeley [1] based on its lack of twisting, smaller articular condyles, and the lack of a ridge connecting ventrally the supposed lateral crest with the supposed humeral head, although he noted that the compression along the anterior margin is similar as in lizards. Despite that, Lydekker (comment to Seeley [1]) suggested a rhynchocephalian relationship of *Saurodesmus robertsoni*. Even though fossils of true Triassic squamates were discovered only relatively recently and remain rare [246–249], Seeley's [1] arguments for rejection of such a relationship remain valid. Triassic lepidosauromorphs still had both the entepicondylar and ectepicondylar foramen, the loss of which is a derived character [247–250]. Both foramina are still retained in *Sphenodon punctatum*, the humerus of which further differs from NHMUK PV OR 28877, e.g., in a much stronger twist, straighter shaft, flatter and more platelike distal expansion, and restriction of the distal articular structures to the midsection of the distal expansion (Fig 7). The rhynchocephalian femur differs in a much lesser expansion of both ends and predominantly ventral direction of the minor trochanter (Fig 7).

## Synapsida

Possible synapsid affinities of NHMUK PV OR 28877 were brought up several times in the 19th century, but only in passing and their significance is in most cases uncertain, given, e.g., the convoluted history and unestablished contents of the group at the time, and the then unresolved problems concerning distinction between the synapsids, parareptiles, and placodonts. Seeley [1] indicated some similarity of the proximal end to the Jurassic mammalian femur described by him earlier [251] but noted dissimilarity of the distal ends; later, he highlighted a similarity of either the distal end [22–24] or the whole bone [26] to dinocephalian femora but also remarked that in *Saurodesmus robertsoni* the proximal and distal ends are more expanded,

the anterior edge of the distal end is more compressed, and the trochanter minor is directed more medially and larger [26]; even later [27], he stated that NHMUK PV OR 28877 as a whole can be compared to the proximal part (the only known at the time) of the femur of his newly described cynodont, *Cynognathus crateronotus*, but he also indicated the ventral rather than anterior inclination of the trochanter minor in the latter and did not declare whether he considered the two taxa to be closely related. Huene [28] suggested, without providing any morphological evidence, its inclusion into his Lycosuchia, containing non-cynodont synapsids but also some problematic taxa, such as *Actiosaurus gaudryi* Sauvage, 1883 [252] and *Crurosaurus problematicus* Huene, 1902 [28]. Both of the latter turned out not to be synapsids, the former being eventually reinterpreted as a choristodere (see Supplementary Information), and the latter remaining problematic [253, 254].

Most historically cited resemblances of NHMUK PV OR 28877 with synapsids outside of the cynodont grade seem to relate to plesiomorphic characters, reflecting a generalized synapsid condition. This is certainly true for dinocephalian femora which share with *Saurodesmus robertsoni* a shallow intertrochanteric fossa and anteroposterior flattening of the whole bone, but otherwise the similarity is rather faint–the femora in dinocephalians are much stockier, lack the anterior deflection of the proximal end, the proximal and distal ends are less flared, the internal trochanter is buttonlike or ridgelike and directed posteriorly, and the anterior and posterior surfaces of the shaft bear a sublongitudinal concavity and may be crossed by a wide ridge connecting one of the condyles with the proximal end (Fig 8) [26, 255–257]. Although the femora of therocephalian synapsids are slenderer than those of dinocephalians and they may exhibit a similar form factor of the distal articular surface to that of NHMUK PV OR 28877, they also are characterized by less flared proximal and distal ends, shallower intertrochanteric fossa, less prominent trochanters, and straighter and ridged shafts [258].

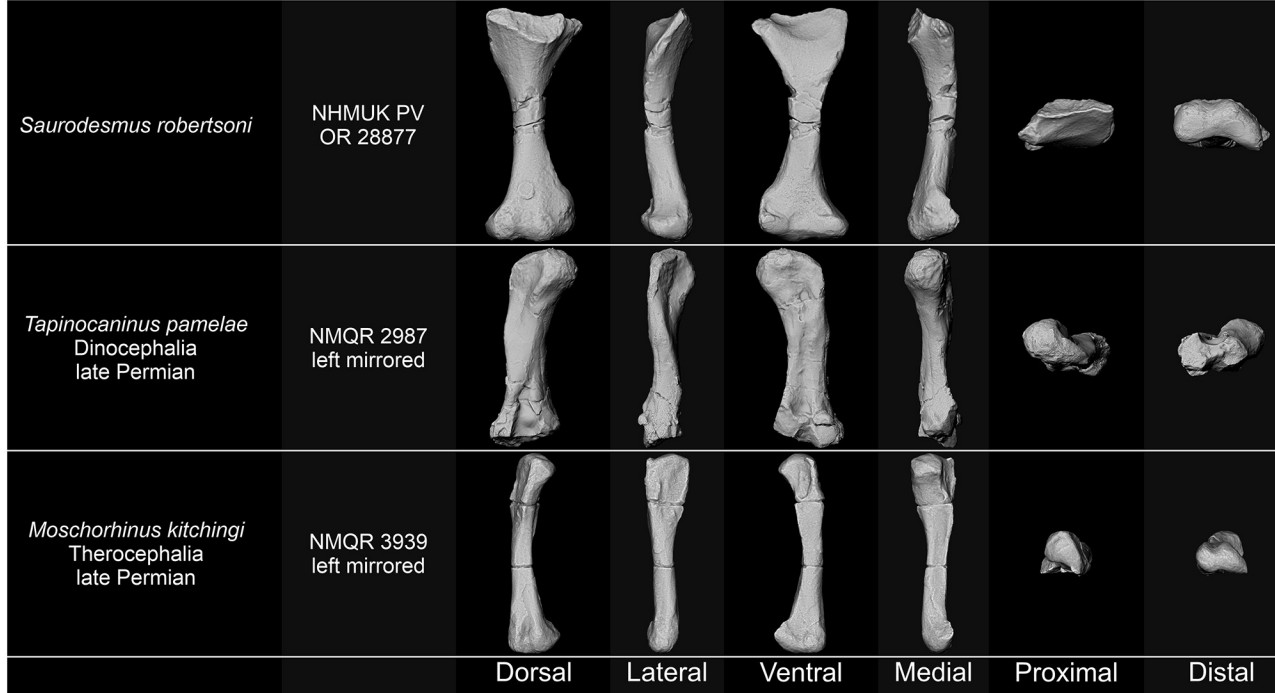

**Fig 8. Comparison of *Saurodersmus robertsoni* NHMUK PV OR 28877 with femora of non-cynodontian synapsids.** Not to scale.

NHMUK PV OR 28877 differs from cynodont humeri in its general form and observable structures, most notably it lacks the rotation between the proximal and distal regions, the deltopectoral crest, the entepicondylar foramen, proportionally mediolaterally expanded distal region, and close alignment of rounded distal condyles clearly separated from the epicondyles, but fits the morphology of cynodont femora (Figs 9–11). While proximally the general outline of the bone and the ventral fossa are indeed similar to Seeley's [251] femur of a Jurassic

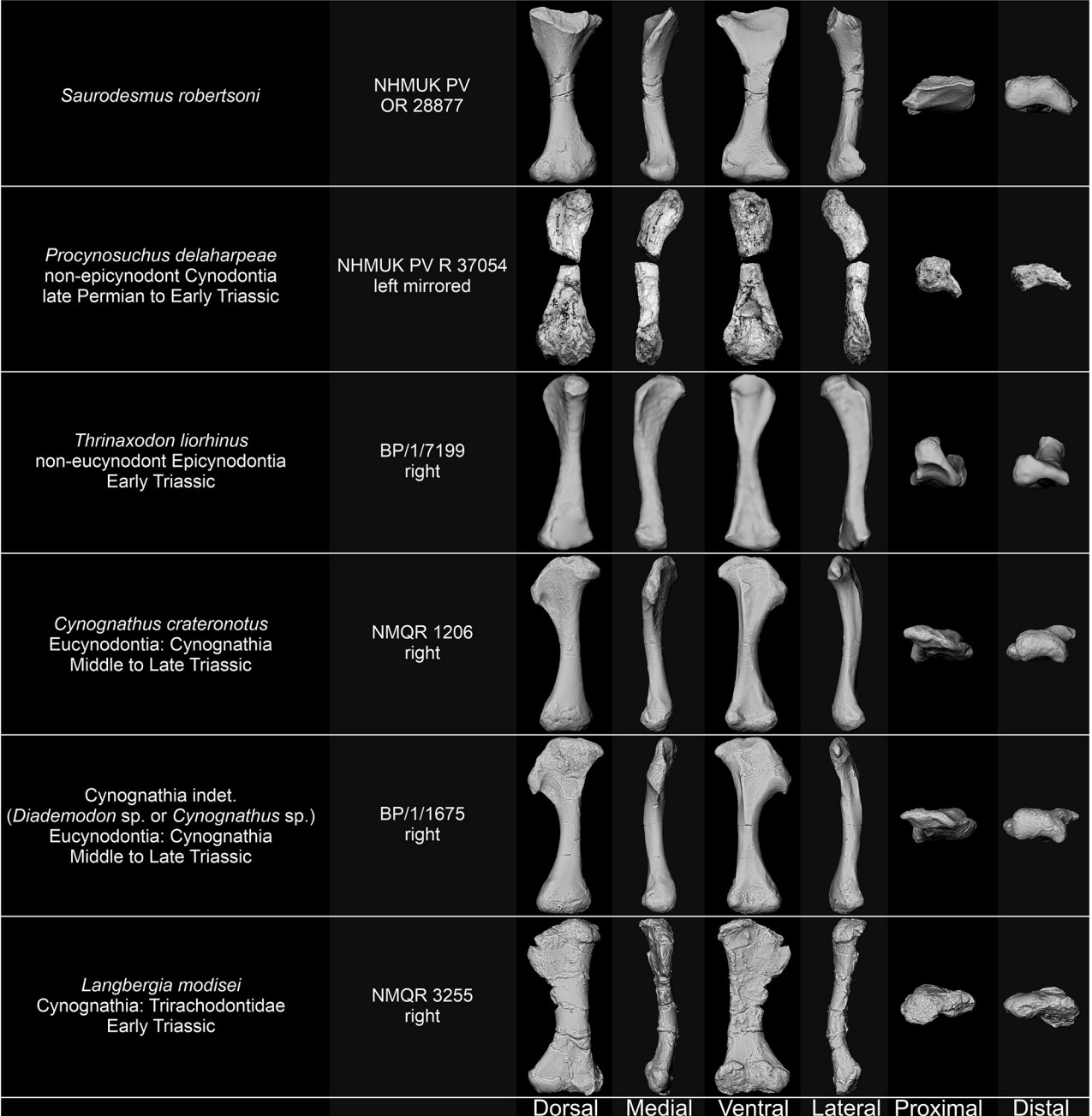

**Fig 9. Comparison of *Saurodersmus robertsoni* NHMUK PV OR 28877 with femora of non-probainognathian cynodonts.** Not to scale. Note the variable development of the hooked greater trochanter in cynognathians.

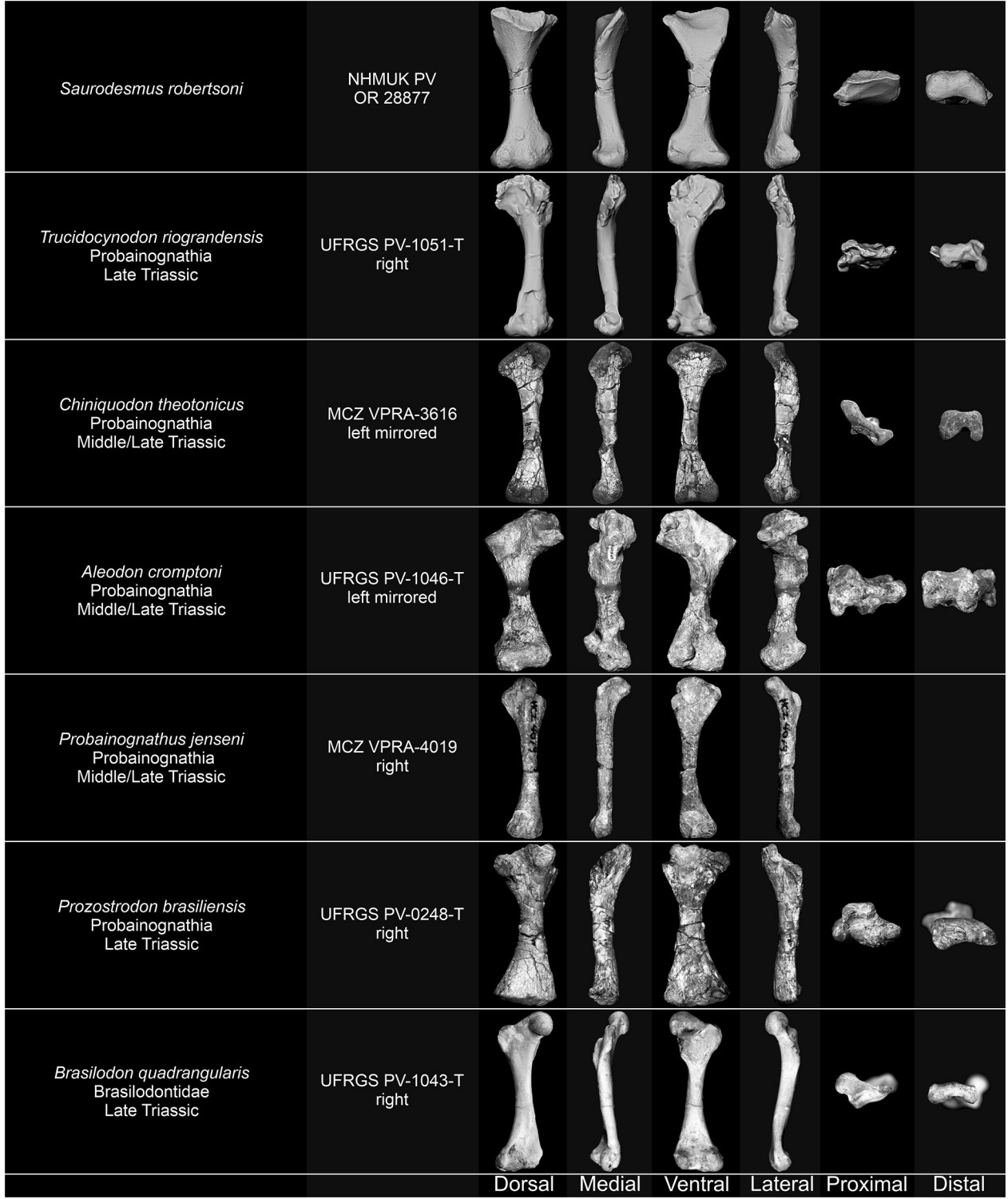

**Fig 10. Comparison of *Saurodersmus robertsoni* NHMUK PV OR 28877 with femora of non-tritylodontid probainognathian cynodonts.** Not to scale.

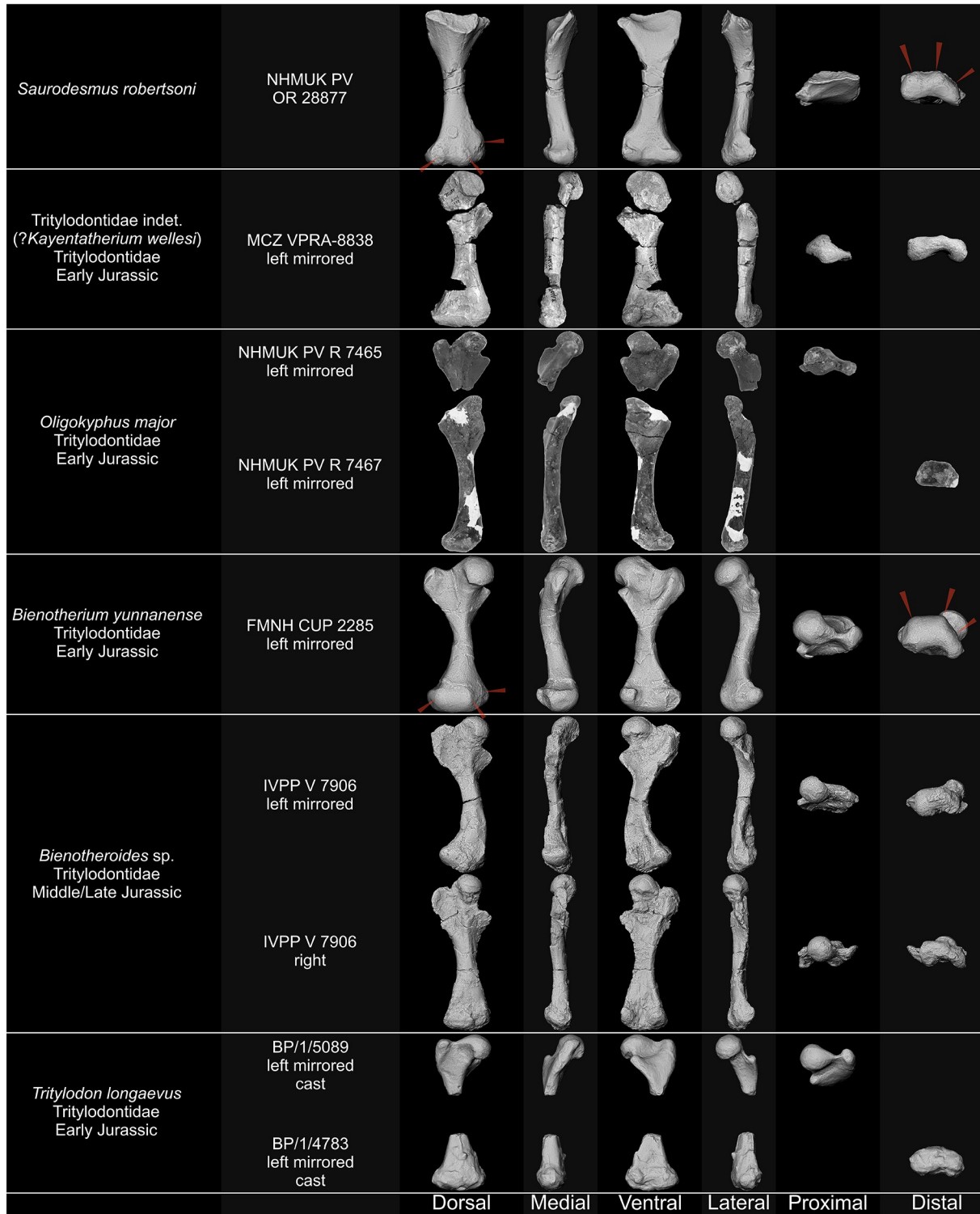

**Fig 11. Comparison of *Saurodersmus robertsoni* NHMUK PV OR 28877 with femora of tritylodontid cynodonts.** Three dorsal eminences proximal to condyles and the epicondyle shared by *Saurodesmus robertsoni* and *Bienotherium yunnanense* indicated by red arrowheads. Note that the middle eminence is located nearly at the level of the long axis of the bone. Not to scale.

mammal from the Stonesfield Slate, we agree with his [1] notion that the more symmetric and less mediolaterally expanded distal end with a more pronounced popliteal fossa, as well as the grooved posterior surface of the shaft are different. A much better comparison can be found with non-mammalian representatives of the Cynodontia.

The anatomy of the femur of *Saurodesmus robertsoni* is comparable to that of many non-mammaliaform cynodonts and early mammaliaforms (i.e., morganucodontans). As stated by Guignard *et al.* [79], some of the variability highlighted below may result from different ontogenetic stages, and the perceived size, shape, and extent of the different structures may be impacted by preservation. Nonetheless, *Saurodesmus robertsoni* displays a fundamentally cynodontian femoral structure.

As observed in dorsal/ventral view, the relative mediolateral expansion of the proximal and distal regions of *Saurodesmus robertsoni* femur are similar to *Bienotheirum yunnanense* (CUP 2285; Fig 11), *Prozostrodon brasiliensis* (Barbarena, Bonaparte, & Teixeira, 1987) [136] (Fig 10), the indeterminate tritylodontid MCZ VPRA-8838 (Fig 11), *Langbergia modisei* (NMQR 3255; Fig 9), and *Traversodon stahleckeri* Huene, 1936 [88], differing from other cynodonts. Among these, the femoral proportions of *Saurodesmus robertsoni* are most similar to those of *Bienotheirum yunnanense* (Fig 11).

The femoral head in NHMUK PV OR 28877 is interpreted to have been offset from the long axis of the shaft, in a degree comparable to that observed in some traversodontids (i.e., *Boreogomphodon jeffersoni* Sues & Olsen, 1990 [123], *Massetognathus ochagaviae* Barberena, 1981 [127], *Luangwa drysdalli* Brink, 1963 [126], *Traversodon stahleckeri*), probainognathians (i.e., *Probainognathus jenseni* Romer, 1970 [135] (Fig 10), *Brasilodon quadrangularis* (Fig 10), *Bienotherium yunnanense* (Fig 11), tritylodontid indet. CXPM C2019 2A235), and morganucodontans (Morganucodontidae indet. PIN 4774/1, *Eozostrodon parvus* Parrington, 1941 [140]). On the other hand, most non-mammalian cynodonts show a less dorsally projected femoral head, including early and derived forms.

The lesser trochanter is medially and slightly ventrally projected, and dorsally exposed, separated from the femoral head by a well-defined notch as in the traversodontid *Andescynodon mendozensis* Bonaparte, 1969 [122], the early probainognathian *Trucidocynodon riograndensis* Oliveira, Soares, & Schultz, 2010 [99] (Fig 10), tritylodontids (i.e., *Tritylodon longaevus*, the unnamed tritylodontid CXPM C2019 2A235 from Lufeng, *Bienotherium yunnanense* (FMNH CUP 2285, FMNH CUP 2295), *Bienotheroides* sp. IVPP V 7906, *Oligokyphus major* Kühne, 1956 [105], the indeterminate tritylodontid MCZ VPRA-8838; Fig 11), *Eozostrodon parvus*, *Morganucodon watsoni* Kühne, 1949 [144], the indeterminate morganucodontid PIN 4774/1 from Peski, and *Haldanodon exspectatus* Kühne & Krusat, 1972 [142]. Among these taxa, *Saurodesmus robertsoni* shares with *Bienotherium yunnanense* (FMNH CUP 2285, FMNH CUP 2295), *Tritylodon longaevus*, *Trucidocynodon riograndensis*, *Oligokyphus major*, *Eozostrodon parvus*, *Morganucodon watsoni*, the indeterminate morganucodontid PIN 4774/1 from Peski, and *Haldanodon exspectatus* the relatively proximal location of the lesser trochanter (Figs 10 and 11). A similar orientation of the lesser trochanter is observed in *Irajatherium hernandezi* Martinelli et al., 2005 [95] and *Therioherpeton cargnini* Bonaparte & Barberena, 1975 [137], but in these taxa it is not separated from the head by a notch. Early cynodonts (i.e., *Thrinaxodon liorhinus* Seeley, 1894 [121], *Galesaurus planiceps* Owen, 1860 [119], *Procynosuchus delaharpeae* Broom, 1937 [120]), most cynognathians (i.e., *Diademodon tetragonus* Seeley, 1894 [124], *Cynognathus crateronotus*, *Cricodon metabolus* Crompton, 1955 [81], *Langbergia modisei* (NMQR 3255), *Luangwa drysdalli*, *Pascualgnathus polanskii* Bonaparte, 1966 [129], *Scalenodon angustifrons* (Parrington, 1946) [130], *Traversodon stahleckeri*, *Massetognathus pascuali* Romer, 1967 [128], *Massetognathus ochagaviae*, *Exaeretodon argentinus* (Cabrera, 1943) [125], *Boreogomphodon jeffersoni*, *Santacruzodon hopsoni*, and *Menadon besairei*) and some

probainognathians (i.e., *Chiniquodon theotonicus* Huene, 1936 [88], *Prozostrodon brasiliensis*, *Brasilodon quadrangularis*) differ from *Saurodesmus robertsoni* in the mostly ventral orientation of the lesser trochanter (Figs 9 and 10).

The general outline and relative development of the greater trochanter of *Saurodesmus robertsoni* is most comparable to that of the traversodontid *Boreogomphodon jeffersoni* and the tritylodontids *Oligokyphus major*, *Tritylodon longaevus*, and the unnamed taxon CXPM C2019 2A235 from Lufeng (Fig 11). The greater trochanter of *Saurodesmus robertsoni* is more laterally flaring and has a more expanded base than in early cynodonts (i.e., *Thrinaxodon liorhinus*, *Galesaurus planiceps*, *Procynosuchus delaharpeae*; Fig 9). It flares less laterally and is longer proximodistally than in many probainognathians (Figs 9 and 10) and in most cynognathians (Fig 9), except for *Exaeretodon argentinus*, which shows a comparable proximodistal extension. The greater trochanter of *Bienotherium yunnanense* (FMNH CUP 2285, FMNH CUP 2295; possibly ontogenetically variable, because the flaring is less prominent in the small FMNH CUP 2298) and *Bienotheroides* sp. IVPP V 7906 is proximodistally comparable but more laterally flaring than in *Saurodesmus robertsoni* (Fig 11). Unlike *Saurodesmus robertsoni* and other non-mammalian probainognathians, the greater trochanter is poorly developed in *Irajatherium hernandezi*. *Saurodesmus robertsoni* lacks a hook-like greater trochanter, differing from the traversodontids *Luangwa drysdalli* and *Traversodon stahleckeri*, and some other cynognathians (Fig 9). Although this character is difficult to assess in many taxa based on the available literature, and may be to some extent influenced by diagenesis, at least the preserved base of the greater trochanter in *Saurodesmus robertsoni* is significantly thinner dorsoventrally (more plate-like) than in *Bienotherium yunnanense* (Fig 11). In the latter taxon, it has a distinctly rhomboid cross-section, regardless of the size/ontogenetic age of the individual (FMNH CUP 2285, FMNH CUP 2295, FMNH 2298). A thinner, plate-like greater trochanter is present in *Tritylodon longaevus* (particularly at the base; Fig 11), the unnamed taxon CXPM C2019 2A235 from Lufeng, and *Brasilodon quadrangularis* (Fig 10).

The shallow intertrochanteric and adductor fossae of *Saurodesmus robertsoni* are separated by a faint intertrochanteric crest. However, most non-mammaliaform cynodonts, bear a strong intertrochanteric crest (i.e., *Cynognathus crateronotus*, *Diademodon tetragonus*, *Luangwa drysdalli*, *Andescynodon mendozensis*, *Massetognathus pascuali*, *Pascualgnathus polanskii*, *Traversodon stahleckeri*, *Chiniquodon theotonicus*, *Probainognathus jenseni*, and *Brasilodon quadrangularis*) or show a single proximoventral fossa (i.e., *Procynosuchus delaharpeae*, *Galesaurus planiceps*, *Langbergia modisei* (NMQR 3255 –the specimen is crushed and there is some residue of matrix and glue, but there is no evidence of an intertrochanteric crest), *Boreogomphodon jeffersoni*, *Massetognathus ochagaviae*, *Exaeretodon argentinus*, *Santacruzodon hopsoni*, *Menadon besairei*, *Trucidocynodon riograndensis*, *Prozostrodon brasiliensis*, *Bienotherium yunnanense*, *Bienotheroides* sp. IVPP V 7906, *Oligokyphus major*, *Tritylodon longaevus*, the indeterminate tritylodontid MCZ VPRA-8838, *Eozostrodon parvus*, *Morganucodon watsoni*, the indeterminate morganucodontid PIN 4774/1 from Peski, and *Haldanodon exspectatus*; Figs 9–11).

The general outline of the distal region of the femur of *Saurodesmus robertsoni* as observed dorsally is most comparable to *Prozostrodon brasiliensis*, the indeterminate tritylodontid MCZ VPRA-8838, *Bienotherium yuannanese*, *Oligokyphus major*, and the unnamed tritylodontid CXPM C2019 2A235, but these taxa lack a well-developed medial epicondyle (Figs 10 and 11). However, a subtle medial epicondyle is present in the large *Bienotherium yuannanese* FMNH CUP 2285 (Fig 11). In the unnamed tritylodontid CXPM C2019 2A235, the lateral condyle is much more projected than in *Saurodesmus robertsoni*. Although this character is not always obvious from in the published literature, at least in *Bienotherium yunnanense* (FMNH CUP 2285) and PIN 4774/1, as in *Saurodesmus robertsoni*, the distinct dorsal eminence proximal to

the medial (tibial) condyle is located close to the level of the long axis of the femoral shaft, that is almost in the middle of the width of the distal expansion (Fig 11).

Ventrally, the medial (tibial) condyle together with the presence of a small, pocket-like fossa proximally to it and walled medially by a sharpened ridge in *Saurodesmus robertsoni* (Fig 2F) resemble the morphology in the unnamed tritylodontid CXPM C2019 2A235. This fossa is also present in *Bienotherium yunnanense* FMNH CUP 2285 (Fig 11), *Brasilodon quadrangularis* (Fig 10), and in the indeterminate morganucodontid PIN 4774/1 from Peski, in which it was interpreted as an attachment site for *M. gastrocnemius* [112]. However, unlike in *Saurodesmus robertsoni*, in these latter taxa the medial condyle is less developed lateromedially and relatively smaller when compared to the lateral condyle. Kühne [105] reported a possibly similar fossa in *Oligokyphus major* but the exact morphology is not clear from the published drawings and the specimens examined by us (Fig 11). The fossa and lateral ridge seem to be absent in juvenile individuals of *Brasilodon quadrangularis* and in other non-mammaliaform cynodonts. Compared to *Saurodesmus robertsoni*, in ventral view the medial and lateral condyles of *Oligokyphus major* are proportionally higher proximodistally and closer to each other (Fig 11).

The distal end of *Saurodesmus robertsoni* is overall most similar to *Bienotherium yunnanense* (Fig 11), *Prozostrodon brasiliensis* (Fig 10), and the Lufeng tritylodontid CXPM C2019 2A235, even despite the damage of the latter specimen. However, in *Bienotherium yunnanense* (Fig 11) the lateral condyle is more ventrally extended than in *Saurodesmus robertsoni*.

Although we were unable to confirm or refute the presence of a distinct medullary cavity (noted by Seeley [1] and contested by Lydekker's comment published alongside that paper), both possible variants (empty cavity or small medullary region crossed by trabeculae) may occur in cynodont femora [259], making this character inconclusive until a better sampling allowing taxonomic distinctions is obtained.

Femoral characters are scarce in phylogenetic matrices focused on the relationships of non-mammaliaform cynodonts [260, 261]. Considering the generally accepted phylogenetic relationships among these taxa [262, 263], only a few of the femoral features discussed above appear to characterize clades whereas the mapping of the others results in a somewhat spotty distribution, even among purportedly closely related taxa, leading to phylogenetic signals that are difficult to interpret (Fig 12). The presence of a mainly medially oriented lesser trochanter is characteristic of derived probainognathians (i.e., tritheledontids, tritylodontids, brasilodontids, and mammaliaforms) opposite to the condition observed in early cynodonts, cynognathians (except *Andescynodon mendozensis*) and early probainognathians (including *Prozostrodon brasiliensis*). Unlike in other derived probainognathians, the lesser trochanter is not separated from the femoral head in tritheledontids.

A slightly less inclusive probainognathian clade, containing tritylodontids, brasilodontids, and mammaliaforms, is characterized by the presence of a projected femoral head, offset from the long axis of the femoral shaft. This trait is interpreted to have been independently acquired by *Probainognathus* and the traversodontids *Boreogomphodon jeffersoni*, *Luangwa drysdalli*, *Massetognathus*, and *Traversodon stahleckeri*. A thin, plate-like greater trochanter (although absent in *Bienotherium yunnanense*), a distinct dorsal eminence proximal to the medial (tibial) condyle located close to the level of the long axis of the femoral shaft (although absent in *Brasilodon quadrangularis*), and a pocket-like fossa proximally to the medial (tibial) condyle (also present in the morganucodontid PIN 4774/1) could also be characteristic of the clade reuniting tritylodontids, brasilodontids, and mammaliaforms.

Overall, the morphology of the femur of *Saurodesmus robertsoni* best corresponds with that in probainognathians more derived than tritheledontids. In particular, it displays morphological traits that, taken together, most closely resemble those observed in tritylodontids, sharing at least with some forms: the relative mediolateral expansion of the proximal and distal regions

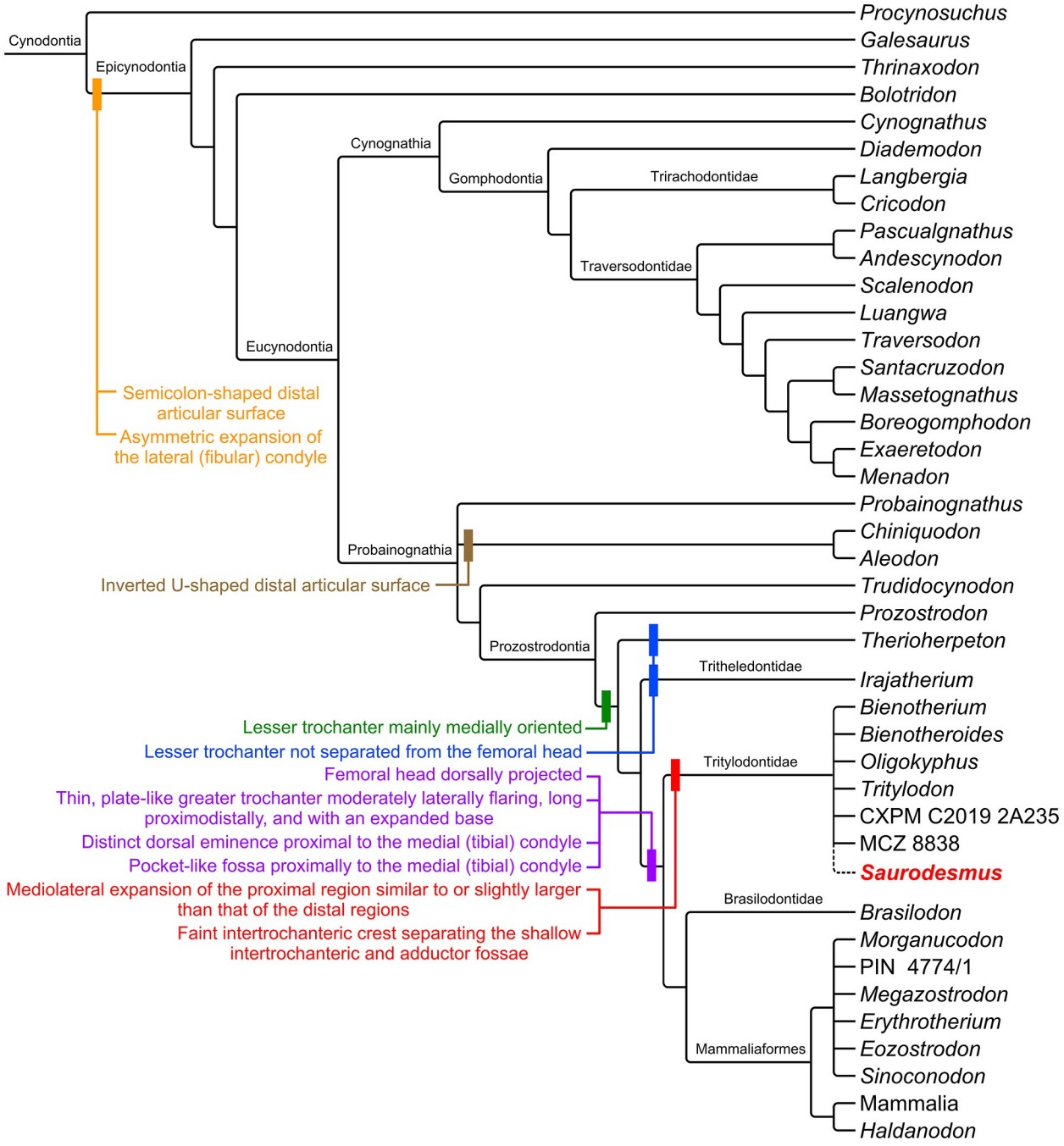

**Fig 12. Cynodont phylogeny (compiled from [264–266], taxa with femora preserved mentioned in the text) with appearances of femoral characters discussed in the text mapped within the Prozostrodontia.** Note that exact distribution of characters is difficult to establish due to scarcity of femoral characters used in phylogenetic analyses and some specimens not referred to the genus level and not included in previous phylogenies.

(with MCZ VPRA-8838); the general shape and development of the greater trochanter (with *Oligokyphus major*, *Tritylodon longaevus*, and CXPM C2019 2A235); the presence of a faint intertrochanteric crest separating the shallow intertrochanteric and adductor fossae (with CXPM C2019 2A235; also shared with *Thrinaxodon liorhinus*); and the general outline of the

distal region as observed dorsally (with MCZ VPRA-8838, *Bienotherium yuannanese*, *Oligokyphus major*, and CXPM C2019 2A235; also similar to *Prozostrodon brasiliensis*) and distally (with *Bienotherium yunnanense* and CXPM C2019 2A235). A well-developed medial epicondyle is unique of *Saurodesmus robertsoni* among tritylodontids.

Although each anatomical feature of NHMUK PV OR 28877 fits within the variability observed in cynodont femora, the combination of characters is unique in comparison with other species with known femoral anatomy, allowing to diagnose *Saurodesmus robertsoni* on the basis of its femoral anatomy. *Saurodesmus robertsoni* is, therefore, the only Triassic cynodont from Scotland. Thus far, the recognized record of Triassic cynodonts in the United Kingdom consisted only of the Mammaliaformes: *Hypsiprymnopsis rhaeticus* Dawkins, 1864 [267] (Allotheria) [267, 268], *Kuehneotherium* sp. (Kuehneotheriidae) [269], *Morganucodon watsoni* (Morganucodontidae) [140], *Thomasia moorei* (Owen, 1871) [268], *Thomasia antiqua* (Plieninger, 1847) [270], and *Thomasia* sp. (Allotheria) [268, 271] from southern England. Only the hypodigm of *Morganucodon watsoni* includes femora, and the direct comparisons on a more taxonomically broad framework performed above show that *Saurodesmus robertsoni* morphology should be interpreted as less derived. If *Saurodesmus robertsoni* is a tritylodontid, it is one of the earliest representatives of that group. Thus far, only two potential Norian representatives of the Tritylodontidae were described from the USA. One of them is a distal humerus identified as Tritylodontidae? indet. [272]; however, the subequal development of its radial and ulnar condyles seems to arguably better fit morganucodontids (compare with Jenkins [109]). The other is represented by associated postcranial remains from the Los Colorados Formation (Argentina), originally tentatively assigned to *Tritylodon* [273]. However, this interpretation was later challenged by Gaetano et al. [106] who suggested that these remains should be better regarded as an undetermined non-mammaliaform cynodont. In the Rhaetian, the record of tritylodontids consists of *Chalepotherium plieningeri* (Ameghino, 1903) [274] and *Oligokyphus triserialis* Hennig, 1922 [275] from Germany [271, 275]. The oldest (and, apparently, only) previously identified tritylodontid from Scotland, *Stereognathus hebridicus* Waldman & Savage, 1972 [276], appeared in the Middle Jurassic (middle Bathonian) of the Isle of Skye [276]. Conversely, if *Saurodersmus robertsoni* is a brasilodontid, it would be the youngest member of that family and its only occurrence from the Triassic Northern Hemisphere–outside of the Ladinian–Norian of Brazil, the other possible record of the Brasilodontidae thus far is restricted to the Early Triassic of India [79, 277–279]. Of course, these interpretations will need to be re-evaluated and validated considering upcoming discoveries of postcranial material of other cynodont groups, as well as possible cranial and dental material, associated to femora, of *Saurodesmus robertsoni* that, hopefully, will be obtained in the future.

## Conclusions

*Saurodesmus robertsoni*, a problematic taxon from the Rhaetian of Scotland based on a single specimen (NHMUK PV OR 28877), is here identified as a derived non-mammaliaform cynodont. Despite its incompleteness, the taxon is here considered valid, making it not only one of the historically oldest finds of that group, but also one of the oldest valid cynodont names. That solves the nearly two centuries-long history of controversy concerning the identity of NHMUK PV OR 28877 and has some bearing on the understanding of the spatiotemporal distribution of cynodonts in the latest Triassic.

## Supporting information

**S1 File. Additional comparisons and references.** 'Younginiformes', Sauropterygia, Choristodera, basal panarchosaurs, 'Protorosauria', Allokotosauria, stem Archosauria,

Kuehneosauridae.
(DOCX)

## Acknowledgments

Michael Day (NHMUK) is thanked for access to specimens NHMUK PV OR 28877 and NHMUK PV R 37054, for the help with digitization, and for the photographs of the femora of *Oligokyphus major*. Gabriela Cisterna (PULR), Eudald Mujal Grané (SMNS), Rainer Schoch (SMNS), Daniela Schwarz (MB), William Simpson (FMNH CUP), and Heike Straebelow (MB) are thanked for granting access to their respective collections. Bernhard Zipfel and Sifelani Jirah (ESI), and Jennifer Botha, Will Archer, Sekhomotso Gubuza, Masabata Chaka, Confidence Nemudivhiso, Nthaopa Nthari, and Thabang Ntsala (NMQR) are thanked for access and repairs to specimens. Caroline Lam is thanked for the access and agreement for reproduction of Jonathan Stiven's illustration of NHMUK PV OR 28877 preserved in the archives of the Geological Society of London. We thank Fernando Abdala for the helpful comments on the manuscript and photographs of *Bienotherium yuannanese* and *Tritylodon longaevus*; Vincent Fernandez (ESRF–European Synchrotron Radiation Facility, Grenoble, France, and ESI) for the synchrotron scan-based 3D model of the femur of *Thrinaxodon liorhinus* BP/1/7199 (the "odd couple" specimen; [280]); Marco Romano (Sapienza University, Rome, Italy, and ESI) for the photogrammetric 3D model of the femur of *Tapinocaninus pamelae* Rubidge, 1991 [281] NMQR 2987 [255]; Justyna Słowiak-Morkovina (ZPAL) for additional photographs (including photogrammetry) of the femora of *Bienotherium yunnanense* (FMNH CUP 2285 and FMNH CUP 2295); Jun Liu and Jicheng Ren (IVPP) for the CT-derived 3D models of both femora or *Bienotheroides* sp. IVPP V 7906; Téo Veiga de Oliveira (Universidade Estadual de Feira de Santana, Feira de Santana, Brasil) for the 3D model of the femur of *Trucidocynodon riograndensis* UFRGS PV-1043-T; Heitor Francischini, Luiz Flávio Lopes, and Laboratório de Paleontologia de Vertebrados (UFRGS) for the photographs of the femora of *Aleodon cromptoni*, *Brasilodon quadrangularis*, *Prozostrodon brasiliensis*, and *Trucidocynodon riograndensis*; Christina Byrd for the photographs of the femora of *Chiniquodon theotonicus*, *Probainognathus jenseni*, and Tritylodontidae indet. (?*Kayentatherium wellesi*). Guilherme Hermanson and colleagues provided access to the *Trionyx triunguis* humerus and femur and *Chelus fimbriata* femur 3D data and Michael Stein and colleagues provided access to the *Crocodilus porosus* humerus 3D data originally appearing in Stein *et al.* [75] with data collection funded by DE150100862, DP140102656, DP140102659, DP130100197, DP170101420, and DP180100792. These files were downloaded from www.MorphoSource.org, Duke University. In Fig 3, silhouettes of the representatives of Aetosauria (CC BY-SA 4.0), Allokotosauria (CC BY-SA 4.0), Anomodontia (CC BY-SA 4.0), Caseasauria (CC BY 2.5), Choristodera (CC BY 2.5), Cynognathia (CC BY-SA 4.0), Dinocephalia (CC BY-SA 4.0), Edaphosauridae (CC BY-SA 4.0), Haramiyida (CC BY-SA 4.0), Kuehneosauridae (CC BY 3.0), Pareiasauria (CC BY 3.0), Sauropterygia (CC BY 2.5), Sphenacodontia (CC BY-SA 4.0) by Nobu Tamura, modified; Aphanosauria (CC BY 3.0), Ornithodira (CC BY-NC-SA 3.0), Phytosauria (CC BY 3.0) by Scott Hartman; Australochelyidae (CC BY 4.0) by Tomasz Szczygielski; Captorhinidae (CC BY-SA 3.0) by Smokeybjb, modified; Crocodylia (CC0 1.0), Rhynchosauria (CC0 1.0) by Steven Traver; Cryptodira by Luca Leicht (PDM 1.0); *Eunotosaurus africanus* (CC BY-SA 3.0), Morganucodonta (CC BY-SA 3.0), *Odontochelys semitestacea* (CC0 1.0), Squamata (CC BY-SA 3.0) by T. Michael Keesey; Euparkeriidae (CC BY 4.0) by Oliver Demuth; Gorgonopsia (CC BY-SA 4.0) by Mario Lanzas, modified; human skeleton (CC0 1.0) by Mariana Ruiz Villarreal, modified; Mammalia (PDM 1.0) by David Orr; *Pappochelys rosinae* (CC BY-SA 3.0) by Rainer Schoch and T. Michael Keesey; Pleurodira (CC0 1.0) by Andy Wilson;

Proganochelyidae (CC BY-SA 4.0) by Ghedoghedo, modified; Proterochersidae (CC BY-SA 4.0) by Conty, modified; 'Protorosauria' (CC BY 4.0) by Fishboy86164577, modified; Rhynchocephalia (CC BY 2.0) by Phillip Capper, modified; Therocephalia (CC BY-SA 3.0) by Dmitry Bogdanov, modified; Tritylodontidae (CC BY-SA 3.0) by Nobu Tamura and T. Michael Keesey; Younginidae (CC BY-SA 3.0) by Miquel Borrull, modified (https://creativecommons.org/publicdomain/mark/1.0/, https://creativecommons.org/publicdomain/zero/1.0/, https://creativecommons.org/licenses/by/2.0/, https://creativecommons.org/licenses/by/2.5, https://creativecommons.org/licenses/by/3.0/, https://creativecommons.org/licenses/by-sa/3.0/, https://creativecommons.org/licenses/by-nc-sa/3.0/, https://creativecommons.org/licenses/by/4.0, https://creativecommons.org/licenses/by-sa/4.0/). The Reviewers, Editor, and editorial team are thanked for their helpful comments and all the work needed to process this manuscript. Opinions expressed and conclusions arrived at, are those of the author and are not necessarily to be attributed to the GENUS.

## Author Contributions

**Conceptualization:** Tomasz Szczygielski.

**Data curation:** Tomasz Szczygielski.

**Formal analysis:** Tomasz Szczygielski, Marc Johan Van den Brandt, Leandro Gaetano, Dawid Dróżdż.

**Funding acquisition:** Tomasz Szczygielski, Marc Johan Van den Brandt.

**Investigation:** Tomasz Szczygielski, Leandro Gaetano, Dawid Dróżdż.

**Methodology:** Tomasz Szczygielski.

**Project administration:** Tomasz Szczygielski.

**Supervision:** Tomasz Szczygielski.

**Validation:** Tomasz Szczygielski.

**Visualization:** Tomasz Szczygielski, Marc Johan Van den Brandt, Dawid Dróżdż.

**Writing – original draft:** Tomasz Szczygielski, Marc Johan Van den Brandt, Dawid Dróżdż.

**Writing – review & editing:** Tomasz Szczygielski, Marc Johan Van den Brandt, Leandro Gaetano, Dawid Dróżdż.

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
