## [Decision Letter · Decision Letter 0]

29 Nov 2023

PONE-D-23-36210Saurodesmus robertsoni Seeley 1891 – the oldest Scottish cynodontPLOS ONE

Dear Dr. Szczygielski,

Thank you for submitting your manuscript to PLOS ONE. After careful consideration, we feel that it has merit but does not fully meet PLOS ONE’s publication criteria as it currently stands. Therefore, we invite you to submit a revised version of the manuscript that addresses the points raised during the review process.

We look forward to receiving your revised manuscript.

Kind regards,

Dawid Surmik, PhD

Academic Editor

PLOS ONE

Journal Requirements:

Additional Editor Comments:

Dear Authors,

Please re-edit the manuscript text based on the reviews you received and respond to them point by point. I look forward to receiving your revised manuscript soon.

David Surmik

Reviewers' comments:

Reviewer's Responses to Questions

**Comments to the Author**

1. Is the manuscript technically sound, and do the data support the conclusions?

Reviewer #1: Partly

Reviewer #2: Yes

2. Has the statistical analysis been performed appropriately and rigorously? 

Reviewer #1: N/A

Reviewer #2: N/A

3. Have the authors made all data underlying the findings in their manuscript fully available?

Reviewer #1: Yes

Reviewer #2: Yes

4. Is the manuscript presented in an intelligible fashion and written in standard English?

Reviewer #1: Yes

Reviewer #2: Yes

5. Review Comments to the Author

Reviewer #1: I agree the bone could belong to tritylodonids, and it may be a valid species, but not clearly shown by current evidences. The authors have exhaustive review on the research history and tried to compare with so many different early tetrapod groups. But for such a bone, I think it is not necessary to do so. It is so different from so many groups, and not need to elaboration, e.g., pareiasaurs. You can explain why the previous identification was wrong by compare with them. However, the comparison within known cynodonts need to be elaboration. Most important, the identification should base on synamorphies. I suggest you summarize diagnostic features on femur for different cynodont groups based on the phylogenetic tree, especially derived, non-mammaliaform prozostrodontians.

This paper do not clearly state why the bone should refer to tritylodontid, and you need more comparison to diagnostic as a valid species within advanced nmc. By the way, the diagnosis need rewrite and add as evidence to support the classification to the abstract.

For the figures 3-7 need to be adjusted. For cynodont femur, you can select one from each group as exemplar, mapped on the cynodont phylogenetic tree (to determine the phylogenetic position). You need detailed comparison with more nmc species to show it is a valid species. You’d better provide more figures of non-mammaliaform prozostrodontians from T3 or J1 such as Oligokyphus.

Abstract : (diagnostic characters)

L41, References begin from 2, need change to [1, 2]

L208 [79-105]

Table 1 list the reference in order

Fig2 Why the color of D-I is different?

L235 in age

Reviewer #2: Dear Editor and Authors,

I am presenting my review of the manuscript Saurodesmus robertsoni Seeley 1891 – the oldest Scottish cynodont by Szczygielski and collaborators.

This is a very interesting manuscript with a strong historical component on one side and detailed morphological component on the other. The historical importance of the material studied is clearly reflected in the introduction. Considering all the comparisons presented by the authors indeed the taxonomical conclusion is accurate and good enough to sustain the taxon as a member of Mammaliamorpha. The implication of this reassignment are nicely presented closing the discussion of the paper.

This paper is indeed very useful and devoted to comparative anatomy in great detail of a long bone. The most commendable is precisely that there are few works nowadays that dedicate in such strong way to comparative anatomy and morphology. The description and comparisons are mostly accurate and accompanied by excellent figures that are extremely useful to follow the text.

The weakness of the manuscript is that several comparisons are not accompanied with figures. That make sense because the comparison is quite broad. Then I wonder if would not be better to reduce the comparisons. In the introduction it is clear that several options were proposed for the taxon. After that, I had the idea that comparison will be mostly with past identities proposed for the taxon, but in the manuscript, I found comparison with taxa that seems to be irrelevant, were not proposed as probable identity for Saurodesmus and, as I said before, are not illustrated. Then my major comment for the manuscript is about this particular point. Perhaps it is possible to reduce the comparative sample?

All in all, the manuscript is a very nice comparative description suitable for publication in PlosOne and my commentaries should be considered as minor changes. An advisable point to consider should be to have images, if possible, of the tritylodontid XPM C2019 2A235, which is presented as the closer match to Saurodesmus and reevaluation of some other images currently in the manuscript. The illustrated materials of Palaeochersis and Langbergia for example appears quite poorly preserved and I suggest to remove them. It is not clear also why is the Ornithorhynchus presented if it is not compared at all in the manuscript.

Best regards,

Fernando Abdala

6. PLOS authors have the option to publish the peer review history of their article (what does this mean?). If published, this will include your full peer review and any attached files.

Reviewer #1: No

Reviewer #2: **Yes: **Fernando Abdala

---

## [Author Response · Author response to Decision Letter 0]

10 Mar 2024

Dear Editor, dear Reviewers,

Thank you for your helpful comments and suggestions. We introduced modifications to the text and indicated them with change tracking. Following Reviewer’s suggestions, we added two figures showing the general amniote and detailed cynodont phylogeny – we hope these will make it easier to follow the discussion. We also managed to fix some minor errors in the text that were not pointed out by the Reviewers, revamped the comparative figures for increased readability, and added images of the humerus and femur of Sphenodon punctatus. Below are our responses to the points raised by the Reviewers.

Yours sincerely,

Tomasz Szczygielski

Reviewer #1

“I agree the bone could belong to tritylodonids, and it may be a valid species, but not clearly shown by current evidences. The authors have exhaustive review on the research history and tried to compare with so many different early tetrapod groups. But for such a bone, I think it is not necessary to do so. It is so different from so many groups, and not need to elaboration, e.g., pareiasaurs. You can explain why the previous identification was wrong by compare with them.”

The aim was to justify the attribution of Saurodesmus robertsoni to cynodonts and rule out its other affinities. Because both Reviewers agree with our identification of the taxon as a cynodont, we are happy to consider this goal fulfilled. We removed the unnecessary comparisons and kept in the main text the ones related to historically published interpretations. The less relevant comparisons are moved to Supplementary Material.

“However, the comparison within known cynodonts need to be elaboration. Most important, the identification should base on synamorphies. I suggest you summarize diagnostic features on femur for different cynodont groups based on the phylogenetic tree, especially derived, non-mammaliaform prozostrodontians.”

We added a summary at the end of the comparative section to improve that aspect. While the approach based on synapomorphies would be optimal, the problem is that very few femoral characters have been used in cynodont phylogenetic analyses (if any at all) and the sampling of femora is spotty, so it is very difficult to establish with certainty which characters are true synapomorphies, homoplasies, or autapomorphies of individual taxa. Nonetheless, we tried to summarize the states observed in particular clades/grades. This is also summarized visually in our new Fig 12.

“This paper do not clearly state why the bone should refer to tritylodontid, and you need more comparison to diagnostic as a valid species within advanced nmc. By the way, the diagnosis need rewrite and add as evidence to support the classification to the abstract.”

We are not sure what the Reviewer meant exactly when it comes to rewriting of the diagnosis, but we included in it some more characters and added the summary of important characters to the abstract. Although the determination of diagnostic features is somewhat hampered by the incompleteness of Saurodesmus robertsoni, limited sample of other cynodont taxa with known femora, and imperfect preservation of some of them, we hope that the diagnosis we prepared is now sufficient.

“For the figures 3-7 need to be adjusted. For cynodont femur, you can select one from each group as exemplar, mapped on the cynodont phylogenetic tree (to determine the phylogenetic position). You need detailed comparison with more nmc species to show it is a valid species. You’d better provide more figures of non-mammaliaform prozostrodontians from T3 or J1 such as Oligokyphus.”

For our analysis, we considered probably every specimen of cynodont femur published thus far, as well as some unpublished material (Langbergia modisei NMQR 3255; Bienotherium yunnanense FMNH CUP 2285). We asked around for additional 3D models and/or photographs to enhance the figures by addition of more taxa. Finally, we added Fig 12 to better represent the phylogeny of cynodonts and character distribution of their femora. The presented figures will be a reference for future researchers because they depict the femoral anatomy of several taxa, following the same convention of presentation for every one of them, facilitating comparisons. They also are useful to follow the descriptive comparisons in the text.

“Abstract : (diagnostic characters)”

Added.

“L41, References begin from 2, need change to [1, 2]”

Fixed.

“L208 [79-105]

Table 1 list the reference in order”

We modified the references to be more in order, but this is not entirely possible, because some of them appear earlier in the text or appear in several cells in the table.

“Fig2 Why the color of D-I is different?”

Panels 2C–2I utilize the pink Lit Sphere Radiance Scaling shader, while the same views of the bone are used also in the remaining figures with the grayscale Lambertian Radiance Scaling shader to enhance the geometric detail. Because the same views are repeated, we used this as an opportunity to use both shaders to better capture geometry and surface characteristics. The Lit Sphere RS informs better about the inclination of all the surfaces because it produces less harsh shadows, and the directions of the surfaces are indicated by subtle differences in color, which is ideal for larger panels, such as in Fig 2. The Lambertian RS produces higher contrast, which is better for smaller panels in the remaining figures, and allows manual manipulation of the light source but with that shader some details may be lost in the shadows.

“L235 in age”

Please note that the phrase reads “of Rhaetian (Late Triassic) age”.

 

Reviewer #2

“Dear Editor and Authors,

I am presenting my review of the manuscript Saurodesmus robertsoni Seeley 1891 – the oldest Scottish cynodont by Szczygielski and collaborators.

This is a very interesting manuscript with a strong historical component on one side and detailed morphological component on the other. The historical importance of the material studied is clearly reflected in the introduction. Considering all the comparisons presented by the authors indeed the taxonomical conclusion is accurate and good enough to sustain the taxon as a member of Mammaliamorpha. The implication of this reassignment are nicely presented closing the discussion of the paper.

This paper is indeed very useful and devoted to comparative anatomy in great detail of a long bone. The most commendable is precisely that there are few works nowadays that dedicate in such strong way to comparative anatomy and morphology. The description and comparisons are mostly accurate and accompanied by excellent figures that are extremely useful to follow the text.”

Thank you very much!

“The weakness of the manuscript is that several comparisons are not accompanied with figures. That make sense because the comparison is quite broad. Then I wonder if would not be better to reduce the comparisons. In the introduction it is clear that several options were proposed for the taxon. After that, I had the idea that comparison will be mostly with past identities proposed for the taxon, but in the manuscript, I found comparison with taxa that seems to be irrelevant, were not proposed as probable identity for Saurodesmus and, as I said before, are not illustrated. Then my major comment for the manuscript is about this particular point. Perhaps it is possible to reduce the comparative sample?”

Yes. The aim was to justify the attribution of Saurodesmus robertsoni to cynodonts and rule out its other affinities. Because both Reviewers agree with our identification of the taxon as a cynodont, we are happy to consider this goal fulfilled. We removed the unnecessary comparisons and kept the ones related to historically published interpretations. Additionally, we now include two new figures (Fig 3 and Fig 12) to make following the discussion easier.

“All in all, the manuscript is a very nice comparative description suitable for publication in PlosOne and my commentaries should be considered as minor changes. An advisable point to consider should be to have images, if possible, of the tritylodontid XPM C2019 2A235, which is presented as the closer match to Saurodesmus and reevaluation of some other images currently in the manuscript.”

We asked the authors of the paper on XPM C2019 2A235 to share a 3D model of that specimen with us. Unfortunately, the specimen is housed in a remote collection in China, and we could not establish any contact with the staff. We were informed by Prof. Liu from the IVPP that they would likely not respond to e-mails in English, anyways. However, we were able to supplement the figures with a number of other tritylodontid and non-tritylodontid femora.

“The illustrated materials of Palaeochersis and Langbergia for example appears quite poorly preserved and I suggest to remove them.”

Although we agree that their preservation is not ideal, it is not so poor as to render them unusable – they still provide some information about the general morphology, proportions, and variability of stylopodial bones in turtles and cynodonts, respectively. Because we present accurate and faithful 3D models of original bones, we feel that any deficiencies of preservation will be apparent to the readers, and thus the panels will not be misleading, as they could be if these imperfect specimens were used as a basis for reconstructions. For that reason, we would prefer to keep them in, unless the Reviewers or the Editor insist on removing them.

“It is not clear also why is the Ornithorhynchus presented if it is not compared at all in the manuscript.”

Following the reviewer suggestion, we removed Ornithorhynchus from the figure.

“Best regards,

Fernando Abdala”

Thank you. We also followed all the comments in the annotated PDF as closely as possible. In most cases they require no additional comments, but below we respond to some points that need additional explanation or clarification.

Lines 64–66: “confuse. Try to separate or introduce the identification of Owen first and then the ideas of Quenstedt (denied Trionyx or even turtle at all.”

Owen’s identification was already established in the first sentence of the same paragraph (lines 56–57: „The specimen was first officially announced by Owen [5] in his report on British fossil reptiles for the year 1841 as a turtle femur resembling (but not identical with) that of Trionyx spp.”).

Lines 338–340: “are not these morphological arguments?”

Yes, they are! But this is pretty much the only work out of the fourteen cited papers which provides such arguments (that is why we wrote “almost never”). We slightly rephrased one of those sentences to make it clearer.

Lines 371: “for what sentence are these quotes? Because all of them are previous to 1956.”

Yes, they testify that limb bones of Proganochelys quenstedtii were present before 1956 (i.e., the publication year of Huene’s “Paläontologie und Phylogenie der Niederen Tetrapoden”) in German collections. This means that the researchers at that time already had a possibility (at least theoretically) to make comparisons of Saurodesmus robertsoni with actual Triassic turtle limb bones. We realized that due to the journal citation style (numbers instead of names and dates) this reference might have been unclear, so we added a clarification to the text.

Line 584: “There should be a way to avoid comparison with structures and bones that are not homologous according to your interpretation. Here sounds strange to compare structures of the humerus with the femur which is what you think the specimen of Saurodesmus is.

Is there any neccesity to compare the bone with the humerus of Aenigmastropus. I guess that there is no femur preserved in the species?

I commend very much the very exhaustive comparison of the paper wich it is the best part because detailed comparative discussion with relevant taxa. This work very well for me in taxa that were previously proposed as being the possible taxonomic identity of S. robertsoni. This also works very well when is possible to read the text using the great figures you provides.

On the other hand, I don´t think it work well without figures. Also I think that there are some kind of excess of comparison, like if for example this particular case, a recently described new taxon in the Permian and in which you are comparing morphologies of different bones.”

This was done for two reasons: (1) the humeri of panarchosaurs are much more similar to Saurodesmus robertsoni than their femora (that is why Seeley described that bone as a crocodilian humerus in the first place) and (2) the specimen was considered by Seeley to represent a crocodilian, but our current understanding of crocodilians (or archosaurs) is much different from the understanding in the early 1890s. Saurodesmus robertsoni was a very enigmatic taxon that remained unrecognized for more than a century, and for that reason newly discovered taxa (unknown in the 19th and even 20th century) could be considered possible suspects. Therefore, the idea was to justify our identification with a broad-scaled differential analysis, at least in a short form (a couple of sentences in the case of dissimilar taxa). However, as explained above, because the Reviewers consider these comparisons unnecessary, and because they accept our identification of Saurodesmus robertsoni as a cynodont, we remove those parts from the main text. We moved them to Supplementary Material, so they can be referred to by curious readers. A short note explaining that was added to the Material and methods section.

Line 119: “5”

The authorship for Traversodon stahleckeri is already indicated in the previous paragraph (the first time the specific name is mentioned).

Line 900: “just check. I am not sure if there is postcranial description of M. ochagaviae.”

Yes, the postcranium was described in Pavanatto AEB, Müller RT, Da-Rosa ÁAS, Dias-da-Silva S. New information on the postcranial skeleton of Massetognathus ochagaviae Barberena, 1981 (Eucynodontia, Traversodontidae), from the Middle Triassic of Southern Brazil. Hist Biol. 2016;28: 978–989.

Line 910–916: “I would rather will leave here only the taxa with which Saurodesmus is more similar.”

While we could only focus on the taxa we consider the most similar to Saurodesmus robertsoni, we think it is more helpful to present their similarity in the context of a more inclusive and diverse taxonomic sample. In our opinion these comparisons are not harmful to the manuscript and could be useful for future researchers, and the necessity of extensive comparisons with various cynodont taxa was highlighted by Reviewer 1. Therefore, we prefer to keep this and other similar parts, unless advised otherwise by the Editor.

Line 934: “Is it possible to be sure of this? In the figure the preservation seems quite poor.”

The specimen is crushed and there is some remaining rock and glue, but the exposed bone surface appears mostly natural and flat, with no evidence of an intertrochanteric crest, so its presence seems unlikely. Nonetheless, we added a short notice about the preservation to make it clearer.

Lines 948–953: “it is possible to indicate this in any figure?”; “This I can recognize from the figure even when it is not indicated. I will suggest to indicate this in any of the ventral views of Saurodesmus”

Yes, these modifications were introduced to Figs 2 and 9.

Line 975: “It is unfortunated that you reach this conclusion but you do not show figures of that particular specimen.”

We asked the authors of the paper on CXPM C2019 2A235 to share a 3D model of that specimen with us, but unfortunately the specimen turned out to be not available due to the remote location in which it is housed.

Line 984: “this is a paper of a Cretaceous symmetrodont mammals from China. Just make sure that is the one that correspond to this quote.”

These references were moved from elsewhere during the work on the manuscript and are now unnecessary here, so they are removed.

Fig 2: “is this really the lesser trochanter? If yes, I would eliminate of the figure so we are sure thet only the most distal portion of the bone is what we see.”; “medial tibial??”

We fixed the labelling in Fig 2 (the medial condyle was indeed erroneously indicated as lateral).

Caption to Fig 

---

## [Decision Letter · Decision Letter 1]

2 Apr 2024

PONE-D-23-36210R1Saurodesmus robertsoni Seeley 1891 – the oldest Scottish cynodontPLOS ONE

Dear Dr. Szczygielski,

Thank you for submitting your manuscript to PLOS ONE. After careful consideration, we feel that it has merit but does not fully meet PLOS ONE’s publication criteria as it currently stands. Therefore, we invite you to submit a revised version of the manuscript that addresses the points raised during the review process.

We look forward to receiving your revised manuscript.

Kind regards,

Dawid Surmik, PhD

Academic Editor

PLOS ONE

Journal Requirements:

Additional Editor Comments:

Dear Dr. Szczygielski,

Please refer to the Reviewer 2 comments as part of the minor revision and eventually correct the necessary elements in the manuscript before the final decision to accept the manuscript for publication.

Sincerely yours, Dawid Surmik

Reviewers' comments:

Reviewer's Responses to Questions

**Comments to the Author**

1. If the authors have adequately addressed your comments raised in a previous round of review and you feel that this manuscript is now acceptable for publication, you may indicate that here to bypass the “Comments to the Author” section, enter your conflict of interest statement in the “Confidential to Editor” section, and submit your "Accept" recommendation.

Reviewer #1: All comments have been addressed

Reviewer #2: (No Response)

2. Is the manuscript technically sound, and do the data support the conclusions?

Reviewer #1: Yes

Reviewer #2: Yes

3. Has the statistical analysis been performed appropriately and rigorously? 

Reviewer #1: N/A

Reviewer #2: N/A

4. Have the authors made all data underlying the findings in their manuscript fully available?

Reviewer #1: Yes

Reviewer #2: Yes

5. Is the manuscript presented in an intelligible fashion and written in standard English?

Reviewer #1: Yes

Reviewer #2: Yes

6. Review Comments to the Author

**Reviewer #1: **In the last review, I asked the synapomrphies, your fig 12 really give some and it lay the foundation for further studies. I did not read in detail in this round, but am satisfy on current structure. Your previous version give a broad comparison and could be a good reference for the study of limb bones of early tetrapods, but not a good paper itself for your purpose.

**Reviewer #2: **This is a much improved version of the manuscript and there are only a few additional comments to the authors included in the pdf.

7. PLOS authors have the option to publish the peer review history of their article (what does this mean?). If published, this will include your full peer review and any attached files.

Reviewer #1: **Yes: **Jun Liu

Reviewer #2: **Yes: **Fernando Abdala

---

## [Author Response · Author response to Decision Letter 1]

22 Apr 2024

Dear Editor, dear Reviewera,

Thank you once again for your helpful comments and suggestions. We introduced modifications to the text and indicated them with change tracking. We furthermore improved the figures by increasing the visibility of arrowheads, adding a sharper photograph of the specimen and natural cross sections in Fig 2, and adding a femur of Procynosuchus.

Yours sincerely,

Tomasz Szczygielski

Reviewer #1

“In the last review, I asked the synapomrphies, your fig 12 really give some and it lay the foundation for further studies. I did not read in detail in this round, but am satisfy on current structure. Your previous version give a broad comparison and could be a good reference for the study of limb bones of early tetrapods, but not a good paper itself for your purpose.”

Thank you very much! 

Reviewer #2

“This is a much improved version of the manuscript and there are only a few additional comments to the authors included in the pdf.”

Thank you very much for all the corrections!

Figure 3: “not sure if it is for the resolution of the manuscript figure, but this should be of better final quality as it is very difficult to read.”

Yes, this is the fault of review PDF resolution, the image is indeed very compressed but it is much clearer in the original.

Figure 4: “use a sharper red color. It is barely visible in the figure.”

The red is now more striking.

Figure 7: “I did not see in the manuscript that you compare Stagonolepis, Parasuchus and Silesaurus morphologies with that of Saurodesmus. If they are compared in the supplementary material (I did not check!) you should then include these images in that section.”, “Make more visible the arrows.”

We do not refer to those taxa specifically, but they are shown as representatives of the groups discussed in the text: parasuchians (we added a more direct reference to phytosaurs and aetosaurs in line 610 and it is discussed more in lines 625–634) and dinosauromorphs (line 658).

Table 1: “Langbergia modisei?”

The femur of Langbergia modisei was not published, we produced a 3D model from photographs of the original specimen taken by one of us (M.v.d.B.). To make it clear, we also listed numbers of all relevant specimens examined for this work, either personally or from unpublished photographs or 3D models.

Lines 337–340: “This is confuse with the "not only" I keep expecting what else (besides incompletness and odd morphology but the third elements that should be lack of knowledge in vertebrate morphology is somewhat altered in the reasoning. 

In addition why you will say odd morphology if it was compared with several taxa?.”

The intended meaning here is clearer when read together with the subsequent sentences. What is meant here is that correct identification of Saurodesmus robertsoni was hampered by its peculiar morphology and incompleteness, but also by the ongoing dynamic development of vertebrate paleontology. This caused difficulties in proper understanding of relationships of various taxa, their anatomy, anatomical diversity and taxonomy, and made some past referrals lose meaning due to the abandonment of some taxa or outdated definitions.

The fact that Saurodesmus robertsoni was compared to various taxa does not preclude it from being considered odd. Various researchers tried to fit the most similar bones available to them at the time but virtually always differences were noted, and the result was considered uncertain.

Line 477: “In Proganochelys they seems to be parallel.”

Please note that we mean here the proximal view.

Line 582: “Check this- It is confuse to me. You consider Saurodesmus to show a bipartite distal end? That is what i beleive from the figure, but here the idea I get from the writing is the opposite.”

No, the distal end of the femur of Saurodesmus robertsoni and most other cynodonts does not have a typical bipartite shape, with two roughly symmetrical, rounded condyles separated by a groove. Instead, it has three eminences above the condyles, visible in dorsal view, as explained in the text and indicated in Fig 11.

Line 768: “Then why to add this here? You are saying that there is no similarity whatsoever between this bone and that of Saurodesmus and nobody found there were similarieties before. Then I don´t follow why to include all this comparison here.”

We added this sentence to clarify and rule out such a possibility for the readers who may not be particularly familiar with the morphology of the crocodilian line archosaurs. We feel it is justified due to the history of assigning Saurodesmus robertsoni to crocodylians and their relatives.

Lines 799–800: “Check writing, the sentence is not close and it is confuse, because it gives me the impression that you are discussing them in relation to Saurodesmus (which does not have osteoderms).”

This part explains the difference in views on the interrelationships and taxonomy of crocodile-line archosaurs and phytosaurs during Huene’s time, which now obscures somewhat the original meaning of Huene’s attribution. However, we fixed the sentence by removing the unnecessary “at” in the middle.

Line 866: “Not sure what do you mean with "form factor"

We mean the general shape and proportions of the bone. 

Line 943: “any quote here? How do you know it is morganucodontid?”

To avoid repeating the same references over and over again and muddling the text, we grouped all of them in Table 1. The fact that the Peski specimen is a morganucodontid was established by Gambaryan and Averianov (2001, ref. 118).

---

## [Editor Report · Decision Letter 2]

6 May 2024

Saurodesmus robertsoni Seeley 1891 – the oldest Scottish cynodont

PONE-D-23-36210R2

Dear Dr. Szczygielski,

We’re pleased to inform you that your manuscript has been judged scientifically suitable for publication and will be formally accepted for publication once it meets all outstanding technical requirements.

Kind regards,

Dawid Surmik, PhD

Academic Editor

PLOS ONE

Additional Editor Comments (optional):

Dear Authors,

Thank you for all necessary corections and fast reply. Now I am happy to accept your manuscript for publication in PLoS One.

All the best, Dawid Surmik
---

## [Editor Report · Acceptance letter]

15 May 2024

PONE-D-23-36210R2 

PLOS ONE

Dear Dr. Szczygielski, 

I'm pleased to inform you that your manuscript has been deemed suitable for publication in PLOS ONE. Congratulations! Your manuscript is now being handed over to our production team.

Kind regards, 

on behalf of

Dr. Dawid Surmik 

Academic Editor

PLOS ONE